# Unveiling the Backbone-Optimizer Coupling Bias in Visual Representation Learning

## Abstract

This paper delves into the interplay between vision backbones and optimizers, unveiling an inter-dependent phenomenon termed **b**ackbone-**o**ptimizer **c**oupling **b**ias (BOCB). We observe that canonical CNNs, such as VGG and ResNet, exhibit a marked co-dependency with SGD families, while recent architectures like ViTs and ConvNeXt share a tight coupling with the adaptive learning rate ones. We further show that BOCB can be introduced by both optimizers and certain backbone designs and may significantly impact the pre-training and downstream fine-tuning of vision models. Through in-depth empirical analysis, we summarize takeaways on recommended optimizers and insights into robust vision backbone architectures. We hope this work can inspire the community to question long-held assumptions on backbones and optimizers, stimulate further explorations, and thereby contribute to more robust vision systems. The source code and models are publicly available.

## 1 Introduction

The past decade has witnessed rapid progress in computer vision, marked by significant strides in network architectures (He et al., 2016; Dosovitskiy et al., 2021; Yu et al., 2024) and optimizers (Sinha & Griscik, 1971; Kingma & Ba, 2015). From AlexNet (Krizhevsky et al., 2012a) to ConvNeXt (Liu et al., 2022a), the vision community has pushed the boundaries of pre-trained backbones in terms of task-specific accuracy, efficiency (*e.g.*, model parameters and inference speed), and other metrics through heuristic architecture designs. Amidst the buzz, however, the impact of optimizers has been largely overlooked - practitioners often default to established ones without systematic justification. For instance, while AdamW (Loshchilov & Hutter, 2019) has emerged as the de facto choice for training Vision Transformers (ViTs), the generality of such optimizer preferences across backbones remains under-explored. This inquiry becomes particularly important as vision models nowadays are deployed in various real-world applications, where the choice of optimizer can significantly impact model generalization (Woo et al., 2023; Oquab et al., 2023), robustness to distribution shifts (Vishniakov et al., 2023), and adaptability in transfer learning (He et al., 2017; Kirillov et al., 2023). Recent studies have explored adapting Adafactor for efficient training scaling in ViTs (Zhai et al., 2022), comparing SGD and AdamW for vision model fine-tuning (Kumar et al., 2022), and investigating general optimizer designs for transformers (Xiong et al., 2020). Thus, understanding the backbone-optimizer interplay may provide critical insights for enhancing model reliability and facilitating vision backbone design and deployment across diverse practical scenarios.

In this paper, we explore the relationship between vision backbones and optimizers. Our primary focus is threefold: **(i)** Does any identifiable dependency exist between existing vision backbones and widely-used optimizers? **(ii)** If such backbone-optimizer dependencies exist, (how) do they affect the training dynamics and performance of vision models? **(iii)** Can we identify direct connections between these inter-dependencies and specific designs of vision backbone architectures and optimizers?

To answer these questions, we first revisit different categories of existing vision backbones and optimizers as shown in Figures 1, A1, and Section 2. We then conduct extensive experiments where 20 representative backbones are evaluated against 20 optimizers on mainstream vision datasets, including CIFAR-100 (Krizhevsky et al., 2009), ImageNet (Krizhevsky et al., 2012b), and COCO (Lin et al., 2014). As such, we provide the backbone-optimizer benchmark and thereby observe an interesting inter-dependent phenomenon, which we term **b**ackbone-**o**ptimizer **c**oupling **b**ias (BOCB). Table 1 conveys this evidence: classical Convolutional Neural Networks (CNNs) like VGG (Simonyan & Zisserman, 2015) and ResNet (Zhang et al., 2022) exhibit a marked co-dependency with SGD (Sinha & Griscik, 1971). In contrast, modern backbones, such as Vision Transformers (ViTs) (Dosovitskiy et al., 2021) and ConvNeXt (Liu et al., 2022a), perform better when paired with adaptive learning rate

optimizers (Loshchilov & Hutter, 2019) (see blue and gray results in Table 1). While most backbones and optimizers are assumed to be unbiased and well-generalized, our findings appear to question it.

To dig deep into such interplay, thorough empirical analyses are conducted from both backbone and optimizer perspectives. We first examine the performance stability of each backbone with vioplinplots as shown in Figure 3, which offers intuitive insights into their robustness across different optimizers. We then analyze the hyper-parameter robustness of all benchmarked backbone-optimizer pairs in Figure 4,5, as the well-designed ones are expected to have robust hyper-parameter settings. To further elucidate the rationale behind BOCB, we visualize the layer-wise patterns and PL exponent alpha metrics (Martin et al., 2021) of learned parameters to examine how different network architectures influence the parameter space and optimization complexity, hence potentially inducing the BOCB phenomenon. As illustrated in Figures 1, 6, and Appendix D, certain stage-wise (hierarchical or isotropic) and block-wise (heterogeneous or not) designs can significantly affect parameter space and hyper-parameters robustness. Interestingly, we further observe that optimizers also introduce *bias* back to backbones (see Section 4). For example, fine-tuning the AdamW pre-trained backbones with the other ones often leads to significant performance degradation, while this is not present when pre-training the model with SGD variants. Moreover, different pre-training optimizers may even alter the parameter patterns of identical backbones. Overall, our findings suggest that BOCB can be introduced by optimizers and backbones and may significantly impact both the pre-training and downstream fine-tuning of vision models, thus limiting their flexibility and practical applications.

In the following sections, we first provide an overview of vision backbones and optimizers in Section 2. We then present the backbone-optimizer benchmark details and our empirical findings in Section 3. Section 4 offers an in-depth analysis of the rationale behind BOCB and takeaways for recommended optimizers and summarized backbone designs. Our main contributions can be summarized as follows:

- We explore the crucial yet poorly studied backbone-optimizer interplay in visual representation learning, revealing the phenomenon of *backbone-optimizer coupling bias* (BOCB).

- We provide the backbone-optimizer benchmark that encompasses 20 popular vision backbones, from classical CNNs to recent transformer-based architectures, and evaluate their performance against 20 mainstream optimizers on CIFAR-100, ImageNet-1K, and COCO, unveiling the practical limitations introduced by BOCB in both pre-training and transfer learning scenarios.

- From the BOCB perspective, we summarize recommendations of optimizers and insights on more robust vision backbone design. The benchmark results also serve as takeaways for user-friendly deployment. We open-source the code and models for further explorations in the community.

## 2 ROADMAPS OF VISION BACKBONES AND OPTIMIZERS

This section provides an overview of most existing vision backbones and optimizers. We first revisit different networks based on their stage-wise macro design (hierarchical or isotropic), building block structures (heterogeneous or not), and core operators (convolution, self-attention, *etc.*). We then dive into mainstream optimizers, emphasizing their distinctive approaches to learning rate adjustment and gradient handling. This serves two purposes: first, it offers an organized framework for understanding the current landscape; second, it facilitates our subsequent analyses, allowing us to draw connections between experimental results and specific techniques, thereby yielding clear observations and insights.

### 2.1 TAXONOMY OF VISION BACKBONE ARCHITECTURES

**Stage-wise Macro Design.** As shown in Figure 1 and Table A1, the overall framework of existing vision backbones can be categorized into two groups: **(i) Hierarchical architectures:** These models (*e.g.*, VGG (Simonyan & Zisserman, 2015), ResNet (He et al., 2016), MobileNet.V2 (Sandler et al., 2018), and EfficientNet (Tan & Le, 2019)) divide the network into multiple downsizing stages, where each stage consists of stacked building blocks with specific designs (*e.g.*, bottleneck, MetaFormer (Yu et al., 2024) or ConvNeXt (Liu et al., 2022a)) for feature extraction. **(ii) Isotropic architectures:** In contrast, backbones like Vision Transformers (ViTs) (Dosovitskiy et al., 2021) and MLP-Mixers (Tolstikhin et al., 2021) employ an isotropic building block stacking, where stand-alone token and channel mixers (*e.g.*, self-attention and MLP) are proposed to capture long-range dependencies with attention-like operators while enabling token prompting for broader applications.

**Intra-block Micro Design.** The building block structures can also be classified into two paradigms: **(i) Homogeneous structures:** Early CNNs like AlexNet (Krizhevsky et al., 2012a) employed a straightforward approach of interleaving convolutions and pooling layers for feature extraction. A

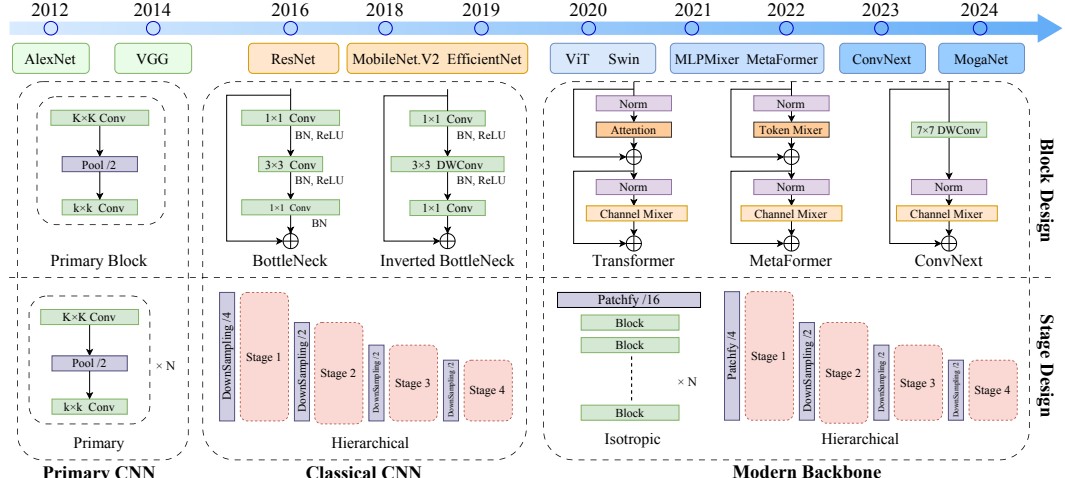

Figure 1: **Vision backbones with representative macro and micro designs** since 2012. **(a) Primary CNNs** like VGG laid the foundation for vision backbone design, *i.e.*, multi-layer networks built by plainly stacking building blocks. **(b) Classical CNNs** like ResNet identified the overall framework of vision backbones as hierarchical stages, each comprising stacked bottlenecks connected by overlapped downsampling layers. **(c) Modern DNNs** introduced different intra-block structures while presenting two main groups of stage-wise design: hierarchical and isotropic stages with downsampling and patchifying. We summarize all the technical details of these typical vision backbones in Table A1.

significant breakthrough came with the bottleneck structure of ResNet (He et al., 2016), which set a new paradigm for subsequent architectures. Following this, research has focused on enhancing bottlenecks through the integration of specialized operators, such as separable convolutions (Sandler et al., 2018) and CBAM (Woo et al., 2018). **(ii) Heterogeneous structures:** ViTs (Dosovitskiy et al., 2021; Liu et al., 2021) marked a paradigm shift by introducing heterogeneous building blocks, in which token-mixers (*e.g.*, self-attention, sliding windows) and channel-mixers (typically feed-forward networks) are exploited for disentangled feature processing. Built upon this, subsequent works mainly focus on crafting more efficient (Yu et al., 2024) and expressive (Ding et al., 2024) token-mixers. Notably, most existing studies change network architectures heuristically to improve certain metrics, such as task accuracy, speed, and parameter efficiency. However, the impact of these architectural choices on their optimization has been largely overlooked, which is exactly what we are interested in.

## 2.2 MAINSTREAM GRADIENT-BASED OPTIMIZERS

The optimization of DNNs is an intricate process requiring iterative parameter updates. Algorithm 1 offers a general framework for this refinement, encapsulating the essence of gradient-based optimizers. The entire pipeline includes four key steps:

**Step 1: Gradient Computation.** The initial phase involves calculating partial derivatives of the loss function $\mathcal{L}$ with respect to each layer's parameters $\theta_l$ through back-propagation. This determines the

---

**Algorithm 1** General Gradient-based Optimizer for DNNs

**Require:** DNN parameters $\theta = \{\theta_l\}_{l=1}^{L}$, an initial learning rate lr, weight decays $\omega = \{\omega_l\}_{l=1}^{L}$, a loss function $\mathcal{L}$, and a dataset $\mathcal{D}$.
1: Initialize parameters $\{\theta_l^0\}_{l=1}^{L}$ and learning rates $\{\alpha_i^0\}_{l=1}^{L} \leftarrow$ lr.
2: **for** each iteration $i = 1, 2, \ldots, L$ **do**     ▷ Loop over iterations
3:     **for** each layer $l = 1, 2, \ldots, L$ **do**     ▷ Loop over layers
4:         Compute gradients $\nabla\theta_l^{i-1} = \frac{\partial\mathcal{L}(\theta,\mathcal{D})}{\partial\theta_l}$.     ▷ Step 1
5:         Estimate gradients $g_l^i$ with $\nabla\theta_l^{i-1}$ and $\{g_l^j\}_{j=1}^{i-1}$.  ▷ Step 2
6:         Caculate $\alpha_l^{i-1}$ with $\{\alpha_l^j\}_{j=1}^{i-1}$ and $\{g_l^j\}_{j=1}^{i}$.     ▷ Step 3
7:         Update: $\theta_l^i \leftarrow \theta_l^{i-1} - \alpha_l^i \cdot \left(g_l^{i-1} + \omega_l \cdot \theta_l^{i-1}\right)$.  ▷ Step 4
8:     **end for**
9: **end for**

---

update direction for each model parameter that can minimize the learning objectives. **Step 2: Gradient Estimation.** To further improve the optimization stability and convergence, gradients can be refined by incorporating both current and historical information. Techniques like momentum are thus employed to smooth gradient estimates, thereby providing more robust and reliable updates. **Step 3: Learning Rate Calculation.** At this stage, the critical hyper-parameter, learning rate, is calculated according to the past statistics and estimated gradients through adaptive optimizers (*e.g.*, AdaGrad, RMSProp, and Adam) for better convergence. **Step 4: Parameter Update.** The final step updates

Figure 2: **Overview of developing timelines for networks and optimizers.** Before the emergence of Transformers, network architectures like ResNet (He et al., 2016) were primarily designed with SGD as the default optimizer. Following the introduction of Transformer (Vaswani et al., 2017) and ViT (Dosovitskiy et al., 2021), AdamW became the standard, accompanied by more sophisticated training strategies, reflecting the increasing complexity of modern architectures.

the network parameters $\theta_l$ with refined gradients $g_l$, learning rates $\alpha_l$, and a weight decay term $\omega_l$ while incorporating additional regularization policies to mitigate overfitting.

As such, mainstream optimizers can be divided into four principal classes as depicted in Figure A1. **(a) Fixed Learning Rate with Momentum:** This category employs static learning rates while modulated by momentum (Sinha & Griscik, 1971; Heo et al., 2021). Its key principle is the accumulation of past gradients to determine the current update, which aids in accelerating convergence in consistent directions while dampening oscillations in high-curvature dimensions. **(d) Adaptive Learning Rate without Momentum:** Optimizers in this class (*e.g.*, AdaGrad (Duchi et al., 2011) and RMSProp (Hinton, 2012)) adapt learning rates for each parameter based on the historical statistics. While this approach may introduce extra cost, it allows for larger updates for infrequent parameters and smaller updates for frequent ones, providing better adaptability to varying feature scales and data sparsity. **(b) Adaptive Learning Rate with Momentum:** This type combines the benefits of momentum with parameter-wise learning rate adaptation (Kingma & Ba, 2015; Reddi et al., 2018), making it well-suited for large datasets or complex neural networks. **(c) Estimated Learning Rate with Momentum:** These optimizers aim to mitigate the convergence issues of Category **(b)** through additional constraints or estimations, such as factored moments (Shazeer & Stern, 2018) and bounded learning rates (Luo et al., 2019).

## 3 BACKBONE-OPTIMIZER COUPLING BIAS (BOCB)

### 3.1 COMBINED EVALUATION OF BACKBONE AND OPTIMIZER

It is commonly assumed that both backbones and optimizers should be broadly applicable and can be combined freely without significant inter-dependence. To investigate the potential backbone-optimizer interplay between a set of vision backbones $\{\mathcal{F}_i(\cdot; \theta)\}_{i=1}^{N_b}$ and widely used optimizers $\{\mathcal{O}_j(\cdot)\}_{j=1}^{N_o}$, we consider three different aspects of evaluation, from task-specific accuracy to optimization dynamics, to identify and then explain the BOCB phenomena (if it exists) with a standard benchmark.

**(A) Performance Metrics.** We assess each backbone-optimizer combination with the top-1 accuracy on the validation set to study whether a backbone relies on (or fails with) the certain optimizer. Given a backbone $\mathcal{F}_i$ and a set of its results $R_i = \{r_j(\mathcal{F}_i)\}_{j=1}^{N_o}$, we detect the failure case that is dynamically lower than others with quantiles and a threshold $\gamma > 0$,

$$r_j(\mathcal{F}_i) < \max(R_i) - \min\big(Q_{0.75}(R_i) - Q_{0.25}(R_i), \gamma\big). \tag{1}$$

Meanwhile, the severity of BOCB can also be reflected by the standard deviation (Std) and range. Therefore, we report these statistics by removing the worst result $\min(R_i)$ and highlight the top-4 results in blue while marking the failed attempts in gray, which yields a heatmap-like table of benchmarking results as a clear overview of the effectiveness of each backbone-optimizer combination.

**(B) Hyper-parameter Robustness.** While standard metrics offer basic insights, we delve deeper into the adaptability of these backbone-optimizer pairs through the lens of hyper-parameter robustness. To quantify this stability, we measure the *variation* of all optimal optimizer hyper-parameters from their mode (most common) configurations. Assuming there are $n$ optimal learning rates and $m$ optimal

weight decays for all backbone-optimizer pairs, we convert these values into discrete one-hot codes $\{\tilde{\text{lr}}_i\}_{i=1}^{n}$ and $\{\tilde{\omega}_i\}_{i=1}^{m}$, calculate mode statistics $M_{\text{lr}}$ and $M_{\omega}$, and measure the *variation* by Manhattan distance $\Sigma_{i=1}^{n}|\tilde{\text{lr}}_i - M_{\text{lr}}| + \Sigma_{i=1}^{m}|\tilde{\omega}_i - M_{\omega}|$. Lower variation often denotes greater robustness of hyper-parameter settings and thereby indicates desirable adaptability to new or poorly studied optimizers or vision backbones and thus could be desirable for broader practical applications.

**(C) Parameter Patterns and Convergence Quality**. To gain intuitive insights into different network architectures, we analyze the learned parameters with four key indicators, including PL exponent alpha (Martin et al., 2021), entropy, $L_2$-norm, and top-$k$ PCA energy ratio. Please view Appendix B.3 for our detailed interpretations. This analysis reveals intrinsic topological patterns that reflect the typical layer-wise properties of various backbones, as shown in Figure 6 and Appendix D. We observe distinct entropy patterns in hierarchical versus isotropic macro designs, variations in $L_2$-norm across stages, and changes in PCA energy ratios for diverse layer types (*e.g.*, convolutional vs. attention-based). By analyzing these patterns, we gain valuable insights into how different network architectures interact with various optimization algorithms, furthering our understanding of the BOCB phenomenon and informing future design choices for both backbones and optimizers.

## 3.2 BENCHMARKS AND OBSERVATIONS

**Benchmark Settings.** We conduct the main benchmark of vision backbones and optimizers for image classification on CIFAR-100 (Krizhevsky et al., 2009) for analysis efficiency, where models are trained 200 epochs with optimal hyper-parameters for various optimizers. We select 20 representative vision backbones from the **three** categories with similar parameters counts, as summarized in Table A1: **(a)** Primary CNNs include AlexNet (Alex) (Krizhevsky et al., 2012a) and VGG-13-BN (VGG) (Simonyan & Zisserman, 2015); **(b)** Classical CNNs include ResNet-50 (R) (He et al., 2016), DenseNet-121 (DN) (Huang et al., 2017), MobileNet.V2 (MobV2) (Sandler et al., 2018), EfficientNet-B0 (Eff) (Tan & Le, 2019), and RepVGG (Ding et al., 2021); **(c)** Modern DNNs include Transformers (DeiT-S (Touvron et al., 2021a) and Swin-T (Liu et al., 2021)), MLPMixer-S (MLP) (Tolstikhin et al., 2021), modern CNNs include ConvNeXt-T (CNX) (Liu et al., 2022a), ConvNeXt.V2 (CNXV2) (Woo et al., 2023), MogaNet-S (Moga) (Li et al., 2024), and UniRepLKNet-T (URLK) (Ding et al., 2024). We also evaluate MetaFormer baselines (Yu et al., 2024) with IdentityFormer-S12 (IF), PoolFormerV2-S12 (PFV2), ConvFormer-S12 (CF), and AttentionFormer-S12 (AF), whose only difference is their token-mixer designs. We also selected 20 popular optimizers from the **four** categories as discussed in Figure A1: **(a)** Fixed Learning Rate with Momentum includes SGD-M (Sinha & Griscik, 1971), SGDP (Heo et al., 2021), and LION (Chen et al., 2023); **(b)** Adaptive Learning Rate with Momentum covers Adam (Kingma & Ba, 2015), AdaMax (Kingma & Ba, 2015), AdamP (Heo et al., 2021), AdamW (Loshchilov & Hutter, 2019), LAMB (You et al., 2020), NAdam (Reddi et al., 2018), RAdam (Liu et al., 2020), and Adan (Xie et al., 2023). **(c)** Estimated Learning Rate with Momentum involves AdaBelief (Zhuang et al., 2020), AdaBound (Luo et al., 2019), AdaFactor (Shazeer & Stern, 2018), LARS (Ginsburg et al., 2018), NovoGrad (Ginsburg et al., 2020), and Sophia (Liu et al., 2023); **(d)** Adaptive Learning Rate without Momentum comprises AdaGrad (Duchi et al., 2011), AdaDelta (Zeiler, 2012), and RMSProp (Hinton, 2012). We consider two training recipes: (1) PyTorch-style training (Szegedy et al., 2016) with basic augmentations, (2) DeiT-style training (Touvron et al., 2021a) with advanced augmentations (Cubuk et al., 2020; Zhang et al., 2018) and techniques (Huang et al., 2016). As for **optimizer hyper-parameters**, we first search the two commonly-employed ones (learning rate and weight decay) with NNI toolbox (Microsoft, 2021), *i.e.*, determining the NNI search range manually. Subsequently, we tune other optimizer-specific hyper-parameters (*e.g.*, momentum in SGD and $\beta_2$ in Adam). The average top-1 accuracy over three trials is reported. Please refer to Appendix B.1 for further implementation specifics.

**Observations.** As shown in Table 1, we observed an interesting phenomenon that some popular models (*e.g.*, DeiT-S and ConvNeXt-T) yield bad results with some optimizers (*i.e.*, SGD and LARS). Therefore, we summarize this phenomenon as BOCB, where the performance of a certain visual architecture is strongly coupled with the choice of the optimizer, deviating from the expected independence between network designs and optimization algorithms. In particular, we notice that classical CNNs (*e.g.*, VGG, ResNets, and RepVGG) exhibit a slight coupling with Category **(a)** optimizers but have not encountered evident BOCB. In contrast, modern architectures like ViTs (Dosovitskiy et al., 2021) and ConvNeXt (Liu et al., 2022a) strongly matched with adaptive optimizers in Category **(b)**.

As observed in Figure 3, we assume that such a coupling bias may stem from the increasing complexity of optimization as network architectures evolve. Concretely, modern backbones incorporate complex designs such as advanced token-mixers (*e.g.*, MogaNet and UniRepLKNet) and block-wise heterogeneous structures (*e.g.*, ConvNeXt variants and CAFormer), which shape a more intricate and

Table 1: Top-1 accuracy (%) of representative vision backbones with 20 mainstream optimizers on CIFAR-100. The torch-style training settings are used for AlexNet, VGG-13, R-50 (ResNet-50), DN-121 (DenseNet-121), MobV2 (MobileNet.V2), and RVGG-A1 (RepVGG-A1), while other backbones adopt modern recipes, including Eff-B0 (EfficientNet-B0), ViTs, ConvNeXt variants (CNX-T and CNXV2-T), Moga-S (MogaNet-S), URLK-T (UniRepLKNet-T), and TNX-T (TransNeXt-T). We list MetaFormer S12 variants apart from other modern DNNs as IF-12, PFV2-12, CF-12, AF-12, and CAF-12. The blue and gray features denote the top-4 and trivial results, while others are inliers. Two bottom lines report mean, std, and range on statistics that removed the worst result for each model.

| Optimizer | AlexNet | VGG-13 | R-50 | DN-121 | MobV2 | Eff-B0 | RVGG-A1 | DeiT-S | MLP-S | Swin-T | CNX-T | CNXV2-T | Moga-S | URLK-T | TNX-T | IF-12 | PFV2-12 | CF-12 | AF-12 | CAF-12 |
|---|---|---|---|---|---|---|---|---|---|---|---|---|---|---|---|---|---|---|---|---|
| SGD-M | 66.76 | 77.08 | 78.76 | 78.01 | 77.16 | 79.41 | 75.85 | 63.20 | 72.64 | 78.95 | 60.09 | 82.25 | 75.93 | 82.75 | 86.21 | 77.40 | 77.70 | 83.46 | 83.02 | 81.21 |
| SGDP | 66.54 | 77.56 | 79.25 | 78.93 | 77.32 | 79.55 | 75.26 | 63.53 | 69.24 | 80.56 | 61.25 | 82.43 | 80.86 | 82.18 | 86.12 | 77.55 | 77.53 | 83.54 | 82.88 | 81.56 |
| LION | 62.11 | 73.87 | 75.28 | 75.42 | 74.62 | 76.97 | 73.55 | 74.57 | 74.19 | 81.84 | 82.29 | 82.53 | 85.03 | 83.43 | 86.96 | 78.65 | 79.66 | 84.62 | 82.41 | 79.59 |
| Adam | 65.29 | 73.41 | 74.55 | 76.78 | 74.56 | 76.48 | 75.06 | 71.04 | 72.84 | 80.71 | 82.03 | 82.66 | 84.92 | 84.73 | 86.23 | 78.39 | 79.18 | 84.81 | 81.54 | 82.18 |
| Adamax | 67.30 | 73.80 | 75.21 | 73.52 | 74.60 | 78.37 | 74.33 | 73.31 | 73.07 | 81.28 | 80.25 | 81.90 | 84.51 | 83.81 | 86.34 | 78.02 | 79.55 | 84.31 | 81.83 | 82.50 |
| NAdam | 60.49 | 73.96 | 74.82 | 76.10 | 75.08 | 77.06 | 74.86 | 72.75 | 73.77 | 81.80 | 82.26 | 82.72 | 85.23 | 82.07 | 86.44 | 78.37 | 80.32 | 84.81 | 81.82 | 82.83 |
| AdamW | 62.71 | 73.90 | 75.56 | 78.14 | 76.88 | 78.77 | 75.35 | 72.15 | 73.59 | 81.30 | 83.52 | 83.59 | 86.19 | 86.30 | 87.51 | 79.39 | 80.55 | 85.46 | 82.24 | 83.60 |
| LAMB | 66.90 | 75.55 | 77.19 | 78.81 | 77.59 | 78.77 | 77.04 | 75.39 | 74.98 | 83.47 | 84.13 | 84.93 | 86.04 | 84.99 | 87.37 | 80.21 | 80.01 | 85.40 | 83.16 | 83.74 |
| RAdam | 61.69 | 74.64 | 75.19 | 76.40 | 75.94 | 77.08 | 74.83 | 72.41 | 72.11 | 79.84 | 82.18 | 82.69 | 84.95 | 84.26 | 86.49 | 78.46 | 79.71 | 84.93 | 81.44 | 82.35 |
| AdamP | 60.27 | 75.56 | 78.17 | 78.89 | 77.79 | 78.65 | 77.67 | 71.55 | 73.66 | 80.91 | 84.47 | 84.40 | 86.45 | 86.19 | 87.16 | 79.20 | 81.70 | 85.15 | 82.12 | 83.40 |
| Adan | 63.98 | 74.90 | 77.08 | 79.33 | 77.73 | 78.43 | 76.99 | 76.33 | 74.94 | 83.35 | 84.65 | 84.77 | 86.46 | 86.75 | 87.47 | 80.59 | 83.23 | 85.58 | 83.51 | 84.89 |
| AdaBound | 66.59 | 77.00 | 78.11 | 75.26 | 78.76 | 79.88 | 74.14 | 68.59 | 70.31 | 80.67 | 71.96 | 83.90 | 78.48 | 83.03 | 86.07 | 77.99 | 77.81 | 82.73 | 83.08 | 82.38 |
| LARS | 64.35 | 75.71 | 78.25 | 77.25 | 76.23 | 72.43 | 75.50 | 71.36 | 72.64 | 81.29 | 61.40 | 82.22 | 33.26 | 41.03 | 85.16 | 77.66 | 78.78 | 82.98 | 81.00 | 82.05 |
| AdaFactor | 63.91 | 74.49 | 75.41 | 77.03 | 75.38 | 77.83 | 75.03 | 74.02 | 71.16 | 80.36 | 82.82 | 83.06 | 85.17 | 85.99 | 86.57 | 78.78 | 78.81 | 84.90 | 81.94 | 82.36 |
| AdaBelief | 62.98 | 75.09 | 80.53 | 79.26 | 75.78 | 78.48 | 76.90 | 70.66 | 73.30 | 80.98 | 83.31 | 84.47 | 84.80 | 84.54 | 86.64 | 78.55 | 81.01 | 85.03 | 83.21 | 83.56 |
| NovoGrad | 64.24 | 76.09 | 79.36 | 77.25 | 71.26 | 74.23 | 75.16 | 73.13 | 67.03 | 81.82 | 79.99 | 82.01 | 82.96 | 80.77 | 85.85 | 77.16 | 78.92 | 83.51 | 81.28 | 82.98 |
| Sophia | 64.30 | 74.18 | 75.19 | 77.91 | 76.60 | 78.95 | 75.85 | 71.47 | 72.74 | 80.61 | 83.76 | 83.94 | 85.39 | 84.20 | 86.60 | 77.67 | 78.90 | 84.58 | 81.67 | 82.96 |
| AdaGrad | 45.79 | 71.29 | 73.30 | 51.70 | 33.87 | 77.93 | 46.06 | 67.24 | 67.50 | 75.83 | 75.63 | 50.34 | 83.03 | 82.57 | 66.83 | 44.34 | 44.40 | 79.67 | 78.71 | 38.09 |
| AdaDelta | 66.87 | 74.14 | 75.07 | 76.82 | 75.32 | 77.88 | 74.58 | 65.44 | 71.32 | 80.25 | 74.25 | 82.74 | 81.06 | 84.17 | 85.31 | 75.91 | 76.40 | 84.05 | 82.62 | 82.08 |
| RMSProp | 59.33 | 73.30 | 74.25 | 75.45 | 73.94 | 76.83 | 74.92 | 70.71 | 71.63 | 77.52 | 82.29 | 82.11 | 85.17 | 61.14 | 86.21 | 77.40 | 77.14 | 84.01 | 79.72 | 81.83 |
| Mean | 63.67 | 74.68 | 76.31 | 76.94 | 75.65 | 77.77 | 75.19 | 70.82 | 72.10 | 80.63 | 78.13 | 82.92 | 83.51 | 82.40 | 86.34 | 78.03 | 78.94 | 84.28 | 81.99 | 82.32 |
| Std/Range | 1.1/8 | 1.0/4 | 1.6/6 | 1.4/6 | 1.6/8 | 1.2/6 | 0.9/4 | 2.9/13 | 1.7/8 | 1.1/6 | 8.0/25 | 0.8/3 | 2.8/11 | 5.5/26 | 0.6/2 | 0.8/5 | 1.2/7 | 0.8/3 | 0.9/4 | 0.9/5 |

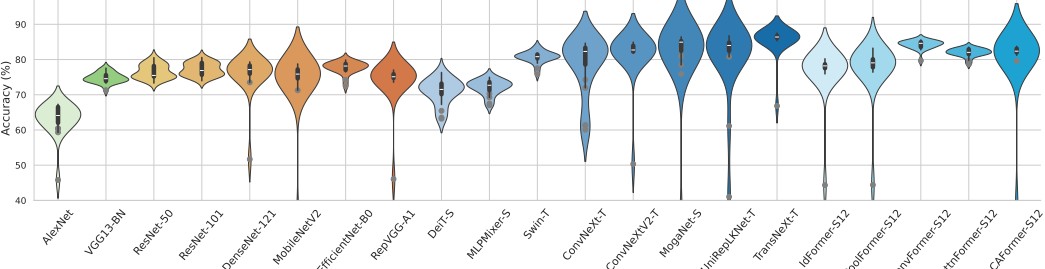

Figure 3: **Violinplot of the performance stability for different backbones.** We visualize the results in Table 1 as violin plots to show the performance stability of different vision backbones. In particular, favorable backbones should not only achieve great performance (high mean accuracy) with few optimizers but yield a small performance variance (a flat distribution without outliers). Note that grey dots denote the outliers (backbone-optimizer combination with poor results), revealing the phenomenon of BOCB. We suggest that well-designed (vision) backbones should exhibit both superior performance and great performance stability across optimizers to mitigate the risk of BOCB.

challenging optimization landscape, necessitating adaptive learning rate strategies. Thus, modern backbones exhibit stronger couplings with optimizers that can navigate these complex landscapes. As we meet real-world challenges, it becomes critical to explore network architectures beyond traditional metrics. Optimizers provide an entry point for this investigation. Intuitively, different network architectures might seemingly affect the optimization landscape, thereby influencing the optimization process. We assume that this interplay between backbones and optimizers may have substantial implications for both pre-training and fine-tuning in practical applications. By examining this relationship, we aim to provide insights that can guide the development of more effective and efficient models for computer vision tasks. The BOCB phenomenon also has several implications for vision backbones in terms of user-friendly deployment and more robust architecture design:

**(A) Deployment.** Vision backbones with weaker BOCB offer greater flexibility and are more user-friendly, especially for practitioners with limited resources for extensive hyper-parameter tuning. However, modern architectures like ViTs and ConvNeXt, which exhibit strong coupling with adaptive optimizers, require careful optimizer selection and hyper-parameter tuning for optimal performance.

**(B) Performance and Generalization:** While classical CNNs with weaker coupling offer more user-friendliness, modern DNNs with stronger coupling potentially leads to better performance and

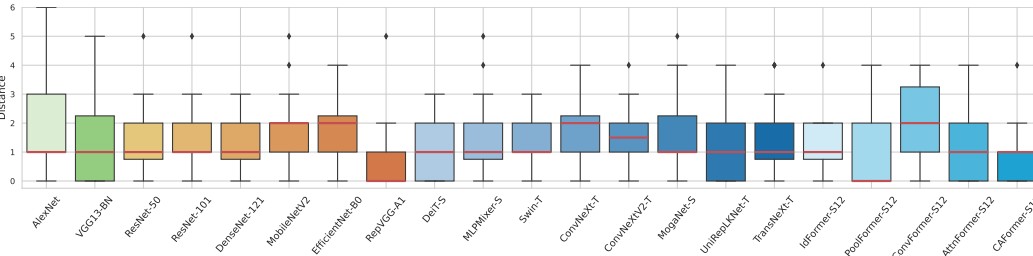

Figure 4: **Boxplot visualization of hyper-parameter robustness** (learning rate and weight decay) for various backbones on CIFAR-100. The vertical axis denotes variation (measured by Manhattan distances) of all optimal hyper-parameters for certain backbones across different optimizers to the default (mode) values. Holistically, backbones with larger mean and variance of variations (*e.g.*, AlexNet, EfficientNet-B0, ConvNeXt-T, and ConvFomer-S12) require more tuning efforts in practice and may be tough to adapt to new or poorly-studied optimizers and tasks. In contrast, models with low variation maximum while the small medians (*e.g.*, ResNet-50, RepVGG-A1, and CAFormer-S12) are regarded as more robust and with more favorable optimization behavior from the view of optimizers.

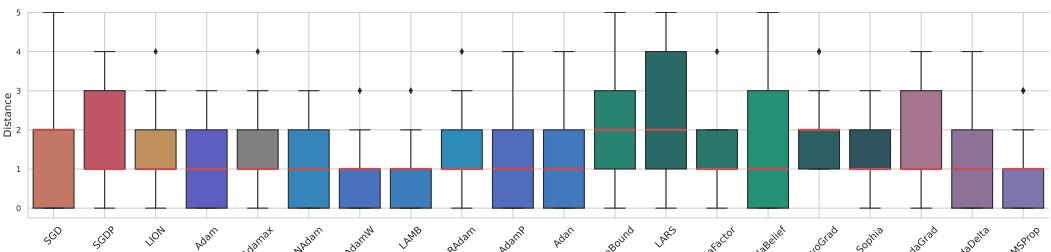

Figure 5: **Boxplot of optimizers generality** across different backbones on CIFAR-100. Symmetrical to Figure 4, the analysis scope here is switched from backbones to optimizers so as to showcase the optimizer's generality from the perspectives of hyper-parameter robustness. Some optimizers in **Category (b)** show favorable robustness (*e.g.*, AdamW and LAMB). Contrastively, several optimizers in the other three types show poor generality (*e.g.*, SGDP, AdaBound, and LARS), which are excluded from our further discussion on the connection between BOCB and diverse vision backbone designs.

generalization. Tailoring the optimization process to certain architectural characteristics of modern backbones, such as hierarchical structures for stage-wise design and depth-wise convolutions for intra-block design for more inductive bias, can effectively navigate complex optimization landscapes, unlocking superior performance and generalization capabilities.

**(C) Backbone Design Insights:** The observed BOCB phenomenon highlights the need to consider the coupling between backbone designs and optimizer choices. When designing new backbone architectures, it is crucial to account for both the inductive bias (*e.g.*, hierarchical structures and local operations) and some optimizing auxiliary modules (Touvron et al., 2021b; Shleifer et al., 2021) introduced by the macro design principles. A balanced approach that harmonizes the backbone design with the appropriate optimizer choice can lead to optimal performance and efficient optimization, enabling the full potential of the proposed architecture to be realized.

## 4 WHERE DOES BOCB COME FROM?

To investigate the essence behind the BOCB phenomenon, we first consider what matters the most: optimizers or backbones. As shown in Figure 5 and Table 1, four groups of optimizers show different extents of BOCB with vision backbones. Categories **(b)** and **(c)** exhibit a robust, hyperparameter-insensitive performance peak, adept at navigating the complex optimization landscapes of early-stage CNNs and recent backbones. Category **(a)** necessitates meticulous hyper-parameter tuning for classical CNNs while demonstrating less adaptability to the high optimization demands of modern backbones with complex designs. Category **(d)** shows the worst performances with heavy BOCB.

### 4.1 ORIGINS OF BOCB: BACKBONE MACRO DESIGN AND TOKEN MIXERS

As discussed in Figure 1 and Section 2, the trajectory of vision backbones has significantly sculpted the optimization landscape, progressing through distinct phases that reflect the intricate relationship between network complexity and training challenges. This section delves into the evolution of vision backbone macro design and its profound implications for the BOCB phenomenon.

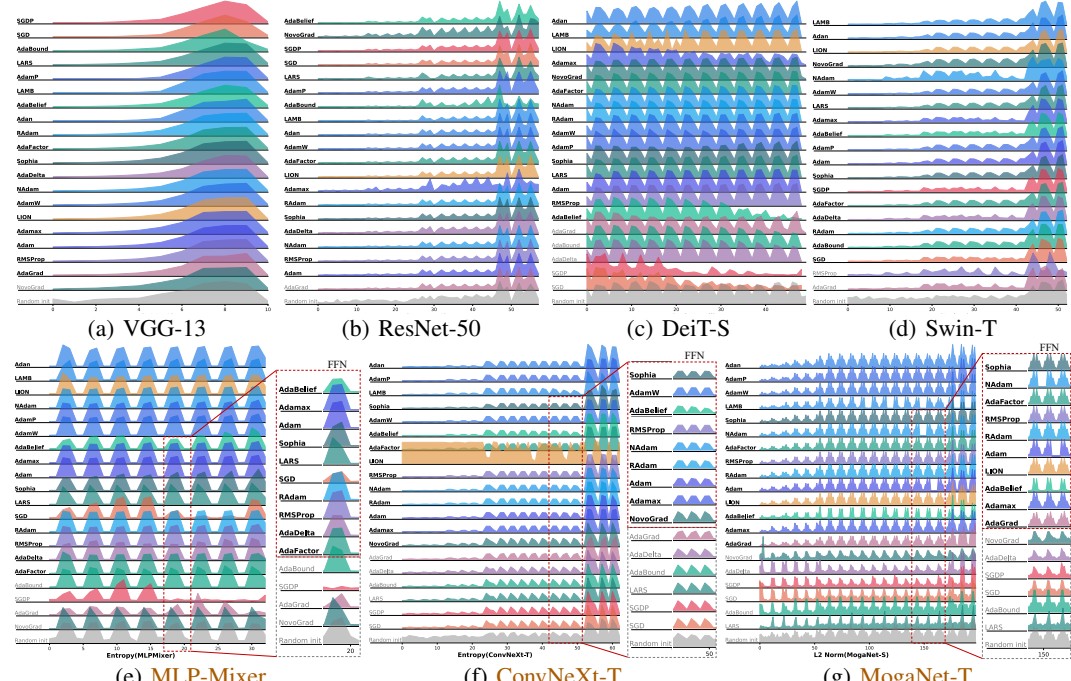

(a) VGG-13          (b) ResNet-50          (c) DeiT-S          (d) Swin-T

(e) MLP-Mixer          (f) ConvNeXt-T          (g) MogaNet-T

Figure 6: **Layer-wise backbone parameter patterns.** We visualize the ridge plot of the entropy patterns of learned parameters of specific vision backbones on CIFAR-100. For each subfigure, the X and Y axes indicate the layer indexes and the entropy of weights, respectively. Specifically, subfigures (a)-(d) represent the ridge plot of the entropy patterns, while subfigures (e)-(g) focus on the ridge plot of the $L_2$-norm patterns for vision backbones with significant BOCB, including MLP-Mixer, ConvNeXt, and MogaNet, *i.e.,* whether the zoomed-in areas of FFN modules are in trivial patterns.

**(i) Early-stage CNNs:** These architectures featured a straightforward design of plainly stacked convolutional and pooling layers, culminated by fully connected layers. Such a paradigm was effective but set the stage for further optimization of landscape alterations. **(ii) Classical CNNs:** The introduction of ResNet marked a pivotal shift towards stage-wise hierarchical designs, significantly enhancing feature extraction and representation learning ability. ResNet-50, in particular, demonstrated a well-balanced approach to BOCB, which exhibited strong compatibility with SGD optimizers and a relatively lower BOCB compared to its contemporaries. **(iii) Modern Architectures:** The transition to modern backbones introduced simplified block-wise designs (*e.g.*, MetaNeXt (Yu et al., 2023) and ConvNeXt variants (Liu et al., 2022a; Woo et al., 2023), or complex block-wise heterogeneous structures (*e.g.*, MogaNet (Li et al., 2024) and UniRepLKNet (Ding et al., 2024)), increasing the optimization challenge and the degree of BOCB due to their complex feature extraction. Representing a pinnacle in evolution, the MetaFormer architecture incorporates both stage-wise and block-wise heterogeneity into its design. This innovative macro design refines the optimization landscape by harmonizing with optimizers, leading to reduced BOCB and enhanced performance.

The above backbone evolution underscores the pivotal role of macro design in shaping the optimization landscape and the necessity for continued innovation in backbone architectures. It highlights the delicate balance that must be struck between advancing representational capabilities and maintaining optimization efficiency. Please view Appendix D for implementation details. Next, we illustrate three cases that elucidate how the overall framework and token mixer designs impact the BOCB phenomena with the parameter quality metric alpha (Martin et al., 2021), demonstrating the representational capacity versus the BOCB effect trade-off.

**Case 1: Transformers.** The lack of inductive biases in ViTs, such as local connectivity and shift-invariance in CNNs, stems from their self-attention mechanism and stage-wise isotropic design. As shown in Figure 7(a), this necessitates careful refinements to ensure effective generalization and reduce BOCB in vision tasks. MLP-Mixer streamlines the model by replacing attention-based token mixers with MLPs, simplifying token interactions and thus inducing a more stable training process. However, it sacrifices the model's capacity to capture long-range dependencies, which is also essential for specific vision tasks, thus representing a trade-off between model simplicity and representation capacity. AttenFormer effectively mitigates BOCB due to its MetaFormer framework, which incorporates balanced designs and residual scaling across stages. Swin-T, akin to DeiT-S, is

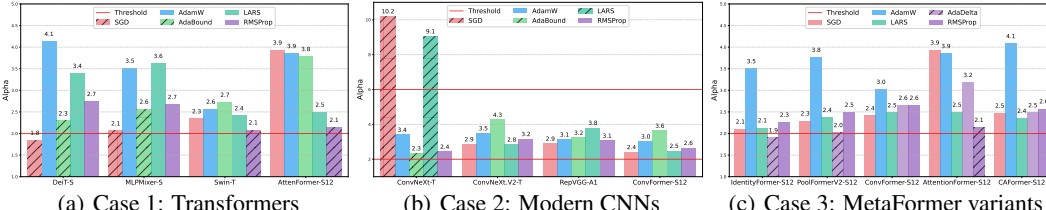

(a) Case 1: Transformers  (b) Case 2: Modern CNNs  (c) Case 3: MetaFormer variants

Figure 7: **BOCB case studies with PL exponent alpha metrics** (Martin et al., 2021) of learned model parameters with diverse optimizers on CIFAR-100. The alpha metric measures the fitting quality of models to a certain task, and a smaller alpha value indicates better fitting. Empirically, values less than 2 or greater than 6 have the risk of overfitting or underfitting. The diagonal bars denote the BOCB occurring. View discussions in Section 4 and details on the alpha metric in Appendix B.3.

based on the Vallina Transformer but introduces hierarchical stages and local attention blocks. These designs enhance the model's ability to capture fine-grained features, leading to better performance and weaker BOCB compared to DeiT-S. *Takeaway: Block-wise macro designs aimed at reducing heterogeneity or enhancing homogeneity, combined with hierarchical stages and the integration of inductive biases within token mixers, are crucial for ViTs to mitigate BOCB in computer vision tasks.*

**Case 2: Modern CNNs.** ConvNeXt, inspired by the macro design of ViTs, introduces a homogeneous block design that integrates two types of operators within a residual connection, potentially enhancing capacities across various tasks and data scales. The effectiveness of this architecture underscores the need to evaluate network designs beyond standard metrics, especially in the context of real-world optimization challenges. The interaction between backbones and optimizers is crucial for both pre-training and fine-tuning, with different architectures influencing optimization landscapes. BOCB in CNNs is often associated with the FFN designs, which are pivotal in models. These blocks, implemented as point-wise convolutions or inverted bottleneck layers, are susceptible to overfitting without proper regularization. To eliminate this, ConvNeXt.V2 introduces Global Response Normalization (GRN) between FFN blocks, similar to RMSNorm, to stabilize training and prevent model collapse, thereby reducing BOCB. ConvFormer, based on the MetaFormer framework, uses homogeneous blocks with depth-wise and point-wise convolutions, improving training robustness and reducing BOCB risk. Similarly, with the VGG series' simple and homogeneous architecture, RepVGG's introduction of training-phase residual connections enhances performance while maintaining stability and weak BOCB (see Figure 7(b)), exhibiting well-behaved training dynamics. In contrast, ConvNeXt and MogaNet, featuring complex operations and heterogeneous blocks, are more susceptible to BOCB. UniRepLKNet, however, sidesteps this issue with a more homogeneous design, highlighting the importance of architectural uniformity in reducing BOCB. *Takeaway: For modern CNNs, designs that foster a homogeneous building block structure and incorporate crafted strategies to mitigate model failures are more likely to achieve stable FFN training and reduce the risk of BOCB.*

**Case 3: MetaFormer.** MetaFormer architecture is distinguished by its hierarchical stage-wise and block-wise design, featuring ResScale, facilitating the flexible integration of various token mixers. This macro design is crucial for achieving competitive performance while minimizing the risk of BOCB. IdentityFormer, without any token mixers, sets a robust baseline for MetaFormer but may fall short in complex tasks requiring advanced token-wise representations, potentially increasing BOCB risk, as shown in Figure 7(c). PoolFormerV2 (pooling as token mixers) outperforms IdentityFormer but may overlook critical details due to the absence of token-wise aggregation, leading to higher BOCB susceptibility. To achieve high performance while mitigating these risks, selecting an appropriate token mixer is essential. ConvFormer integrates CNN layers to balance local inductive bias in data-limited scenarios, ensuring better convergence and less BOCB. AttenFormer and CAFormer further explore attention mechanisms, aiming to enhance the representation capacity with global receptiveness through improved token interactions. *Takeaway: Overall, MetaFormer architecture's success hinges on a judicious balance between its hierarchical design and the selection of token mixers, ensuring robust performance across diverse tasks while mitigating the risk of BOCB.*

## 4.2 PRE-TRAINING AND TRANSFER LEARNING WITH DIFFERENT OPTIMIZERS

**Extending to ImageNet-1K classification.** ImageNet-1K (Krizhevsky et al., 2012b) is a fundamental benchmark that gauges the classification prowess of vision models, and we further investigate whether our observations still hold on ImageNet-1K. View Appendix B.1 for experimental details and Appendix C.2 for extended results. As shown in Table 2, DeiT-S shows stronger BOCB than ResNet-50, while optimizers of **Category (b)** in Figure A1 (*e.g.,* AdamW) have shown a reliable performance peak across diverse backbones during pre-training. Their consistent efficacy is well-aligned with the extensive feature learning required by the ImageNet-1K, making them optimal choices for the initial

Table 2: Top-1 accuracy (%) of DeiT-S and R-50 training 300 epochs by popular optimizers with DeiT and RSB A2 settings on ImageNet-1K.

| Backbone | DeiT-S (DeiT) | R-50 (A2) |
|---|---|---|
| SGD-M | 75.35 | 78.82 |
| SGDP | 76.34 | 78.02 |
| LION | 78.78 | 78.92 |
| Adam | 78.44 | 78.16 |
| Adamax | 77.71 | 78.05 |
| NAdam | 78.26 | 78.97 |
| AdamW | 80.38 | 79.88 |
| LAMB | 80.23 | 79.84 |
| RAdam | 78.54 | 78.75 |
| AdamP | 79.26 | 79.28 |
| Adan | **80.81** | **79.91** |
| AdaBound | 72.96 | 75.37 |
| LARS | 73.18 | 79.66 |
| AdaFactor | 79.98 | 79.36 |
| AdaBelief | 75.32 | 78.25 |
| NovoGrad | 71.26 | 76.83 |
| Sophia | 79.65 | 79.13 |
| AdaGrad | 54.96 | 74.92 |
| AdaDelta | 74.14 | 77.40 |
| RMSProp | 78.03 | 78.04 |

Table 3: Transfer learning to object detection (Det.) with RetinaNet and 2D pose estimation (Pose.) with TopDown on COCO, evaluated by mAP (%) and $AP^{50}$ (%). We employ pre-trained VGG, ResNet-50 (R-50), Swin-T, and ConvNeXt-T (CNX-T) as backbones with different pre-training settings, including 100-epoch (SGD, LARS, or RSB A3), 300-epoch (RSB A2 and Adan), and 600-epoch pre-training (RSB A1).

| Pre-training | 2D Pose Estimation | | | Object Detection | | | | | | | |
|---|---|---|---|---|---|---|---|---|---|---|---|
| | VGG (SGD) | R-50 (SGD) | Swin-T (AdamW) | R-50 (SGD) | R-50 (LARS) | R-50 (A3) | R-50 (A2) | R-50 (A1) | R-50 (Adan) | Swin-T (AdamW) | CNX-T (AdamW) |
| SGD-M | 47.5 | 71.6 | 38.4 | 36.6 | 27.5 | 28.7 | 23.7 | 34.6 | 27.5 | 37.2 | 38.5 |
| SGDP | 47.3 | 41.2 | 38.9 | 36.6 | 17.6 | 18.5 | 26.8 | 26.7 | 27.4 | 37.2 | 22.5 |
| LION | 69.5 | 71.5 | 71.3 | 32.1 | 35.8 | 35.4 | 37.6 | 34.6 | 38.8 | 41.9 | 42.8 |
| Adam | 69.8 | 71.6 | 72.7 | 36.2 | 36.2 | 35.8 | 38.3 | 38.4 | 38.6 | 41.9 | 43.1 |
| Adamax | 69.0 | 71.2 | 72.4 | 36.8 | 36.8 | 36.4 | 38.3 | 38.4 | 38.3 | 41.5 | 42.0 |
| NAdam | 69.7 | 71.8 | 71.9 | 36.0 | 36.6 | 36.1 | 38.2 | 38.4 | 38.7 | 41.9 | **43.4** |
| AdamW | 70.0 | 72.0 | **72.8** | 37.1 | 37.1 | 36.7 | 38.4 | **39.5** | 36.8 | 41.8 | **43.4** |
| LAMB | 68.5 | 71.5 | 71.7 | 36.7 | **37.5** | **37.7** | **38.6** | 38.9 | 38.6 | 41.8 | 42.6 |
| RAdam | 69.8 | 71.8 | 72.6 | 36.6 | 36.5 | 36.0 | 38.2 | 38.4 | 38.6 | 41.6 | 43.3 |
| AdamP | 69.7 | 71.5 | **72.8** | 36.5 | 37.2 | 36.5 | 38.5 | 38.9 | 38.8 | 41.7 | 43.3 |
| Adan | 69.7 | **72.1** | **72.8** | **37.7** | 37.0 | 36.0 | **38.6** | 39.0 | **39.4** | **42.0** | 43.2 |
| AdaBound | 34.0 | 44.9 | 28.4 | 35.9 | 34.2 | 31.9 | 37.0 | 35.0 | 36.7 | 38.8 | 41.2 |
| LARS | 54.4 | 63.4 | 47.6 | 35.8 | 28.9 | 28.8 | 34.7 | 36.9 | 37.3 | 34.6 | 40.5 |
| AdaFactor | **72.8** | 71.7 | 72.7 | 35.6 | 37.0 | 36.4 | 38.5 | 37.8 | 38.7 | 40.5 | 43.1 |
| AdaBelief | 69.6 | 67.0 | 61.8 | 36.2 | 34.4 | 33.1 | 36.4 | 38.2 | 38.5 | 40.0 | 41.4 |
| NovoGrad | 64.2 | 70.7 | 69.8 | 35.6 | 27.2 | 26.3 | 35.2 | 28.6 | 38.5 | 40.4 | 39.0 |
| Sophia | 69.7 | 71.6 | 72.3 | 36.4 | 35.8 | 35.3 | 38.0 | 38.7 | 37.0 | 40.4 | 42.5 |
| AdaGrad | 66.0 | 61.2 | 48.4 | 26.4 | 21.9 | 28.3 | 32.7 | 27.1 | 33.7 | 32.9 | 23.7 |
| AdaDelta | 44.3 | 49.3 | 52.0 | 34.9 | 32.7 | 32.7 | 35.9 | 33.9 | 36.6 | 40.0 | 41.5 |
| RMSProp | 68.8 | 71.6 | 72.5 | 35.3 | 36.2 | 35.6 | 37.8 | 38.3 | 38.7 | 41.5 | 43.1 |

model training phase. Meanwhile, the efficacy of these backbones and optimizers in the pre-training phase cascades to the transfer learning process, as we discussed in the following two paragraphs.

**Transfer Learning on COCO.** As for transfer learning with ImageNet-1K pre-trained models, we have identified two critical findings regarding the performance of COCO object detection (Lin et al., 2017) and 2D pose estimation (Xiao et al., 2018) tasks. Table 3 and Figure A3 provide clear evidence of how various backbones and optimizers perform following transfer from pre-trained models to COCO detection (Lin et al., 2017), indicating *the choice of backbones and optimizers both vital*. From the backbone aspects, the backbone with a pronounced BOCB (ConvNeXt-T) continues to exhibit BOCB characteristics in transfer learning scenarios. This suggests that the inherent structural attributes of such models may not be easily mitigated through transfer learning alone. *Takeaway: The BOCB property is still kept when transferring to dense prediction tasks for pre-trained backbones.*

**Case 4: Optimizer Properties.** We also comprehensively evaluate optimization properties from the view of performance, hyper-parameter robustness, BOCB property, and computational efficiency in Table A5. With transferring experiments shown in Table 3 and Figure 3(b), when we controlled for the BOCB effect in the backbone by using ResNet-50 (less susceptible to BOCB), we observed that optimizers of Category (b) and (c) may introduce significant BOCB effects during the pre-training stage despite their effectiveness in pre-training, indicating that the choice of pre-training optimizer could profoundly influence the generalization and transferring abilities, thereby affecting its transferability and performance on new tasks. Moreover, unlike Category (a), which do not restrict the fine-tuning phase to a specific optimizer, the optimizers in Category (b) and (c) necessitate their use in both pre-training and fine-tuning stages. *Takeaway: Optimizer selection in pre-training can significantly impact models' transferability, with Categories (b) and (c) optimizers potentially introducing BOCB to pre-trained backbones while yielding superior performance. We recommended three superior optimizers and five BOCB indicator optimizers with property evaluation in Appendix D.5.*

## 5 CONCLUSION

This paper explores the interplay of backbone designs and optimizer selections in computer vision. We unveil the phenomenon of **b**ackbone-**o**ptimizer **c**oupling **b**ias (BOCB) and the potential limitations it poses to vision backbones, for example, the extra fine-tuning time and efforts in downstream tasks. We also discover the underlying rationale behind different network designs and BOCB and thereby provide guidelines for future vision backbone design. Meanwhile, the benchmarking results and released code serve as takeaways for user-friendly deployment and evaluation. Overall, we aim to inspire the computer vision community to rethink the relationship between backbones and optimizers, consider BOCB in future studies, and thus contribute to more systematic future advancements.

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

## SUPPLEMENT MATERIAL

The appendix sections are structured as follows:

- In Appendix A, we provide the full description of techniques and categories of the popular vision backbones and optimizers.
- In Appendix B, we provide experimental setups for benchmarks, including information on backbones and optimizers, training recipes, and hyperparameter settings.
- In Appendix C, we provide full experimental results and analysis of the proposed benchmarks.
- In Appendix D, we visualize the learned parameters and explain the BOCB effects.

## A  DETAILS OF POPULAR BACKBONES AND OPTIMIZERS

**Popular Vision Backbones.**   As shown in Table A1, we provide detailed information for 16 typical vision networks with three categories as summarized in Sec. 2.1. The stage-wise and block-wise designs with the optimization techniques of residual branches are regarded as the macro design of DNNs, and the various operators are designed in different networks, which are usually called feature extractors in classical CNNs. The primary and classical CNNs are proposed to use simple training setups (*i.e.*, PyTorch-style setting proposed in Simonyan & Zisserman (2015)) while the modern DNNs have to adopt the complex training recipes like DeiT (Touvron et al., 2021a) and RSB (Wightman et al., 2021) as shown in Table A3.

Table A1: Three categories of typical vision backbones proposed in the past decade. For operators in different network blocks, Conv, SepConv, and DWConv denote normal convolutions, separable convolution, and depth-wise convolution, Gating denotes GLU-like modules (Shazeer, 2020), and SE denotes Squeeze-and-excitation block (Hu et al., 2018). As for the design of residual connection and normalization, the vanilla residual branch use addition (He et al., 2016) or concatenation (Huang et al., 2017), PreNorm denotes the pre-act normalization (Wang et al., 2019) with residual connection, LayerScale (Touvron et al., 2021b) and ResScale (Shleifer et al., 2021) are layer-wise initialization tricks for stabilize training of deep models.

| Backbone | Date | Stage-wise design | Block-wise design | Operator (feature extractor) | Residual branch | Input size | Training setting |
|---|---|---|---|---|---|---|---|
| AlexNet | NeurIPS'2012 | - | Plain | Conv | - | 224 | PyTorch |
| VGG | ICLR'2015 | - | Plain | Conv | - | 224 | PyTorch |
| ResNet | CVPR'2016 | Hierarchical | Bottleneck | Conv | Addition | 32 | PyTorch |
| DenseNet | CVPR'2017 | Hierarchical | Bottleneck | Conv | Concatenation | 32 | PyTorch |
| MobileNet.V2 | CVPR'2018 | Hierarchical | Inv-bottleneck | SepConv | Addition | 224 | PyTorch |
| EfficientNet | ICML'2019 | Hierarchical | Inv-bottleneck | Conv & SE | Addition | 224 | RSB A2 |
| RepVGG | CVPR'2021 | Hierarchical | Inv-bottleneck | Conv | Addition | 224 | PyTorch |
| DeiT-S (ViT) | ICML'2021 | Patchfy & Isotropic | Metaformer | Attention | PreNorm | 224 | DeiT |
| MLP-Mixer | NeurIPS'2021 | Patchfy & Isotropic | Metaformer | MLP | PreNorm | 224 | DeiT |
| Swin Transformer | ICCV'2021 | Patchfy & Hierarchical | Metaformer | Local Attention | PreNorm | 224 | ConvNeXt |
| ConvNeXt | CVPR'2022 | Patchfy & Hierarchical | MetaNeXt | DWConv | PreNorm & LayerScale | 32 | ConvNeXt |
| ConvNeXt.V2 | CVPR'2023 | Patchfy & Hierarchical | MetaNeXt | DWConv | PreNorm & LayerScale | 32 | ConvNeXt |
| MogaNet | ICLR'2024 | Patchfy & Hierarchical | Metaformer | DWConv & Gating | PreNorm & LayerScale | 32 | ConvNeXt |
| UniRepLKNet | CVPR'2024 | Patchfy & Hierarchical | Metaformer | DWConv & SE | PreNorm & LayerScale | 224 | ConvNeXt |
| TransNeXt | CVPR'2024 | Patchfy & Hierarchical | Metaformer | Attention & Gating | PreNorm & LayerScale | 224 | DeiT |
| IdentityFormer | TPAMI'2024 | Patchfy & Hierarchical | Metaformer | Identity | PreNorm & ResScale | 224 | RSB A2 |
| PoolFormerV2 | TPAMI'2024 | Patchfy & Hierarchical | Metaformer | Pooling | PreNorm & ResScale | 224 | RSB A2 |
| ConvFormer | TPAMI'2024 | Patchfy & Hierarchical | Metaformer | SepConv | PreNorm & ResScale | 224 | RSB A2 |
| AttentionFormer | TPAMI'2024 | Patchfy & Hierarchical | Metaformer | Attention | PreNorm & ResScale | 224 | RSB A2 |
| CAFormer | TPAMI'2024 | Patchfy & Hierarchical | Metaformer | SepConv & Attention | PreNorm & ResScale | 224 | RSB A2 |

**Popular Optimizers.**   We also summarize popular optimizers with four categories in Figure A1 and Table A2 provide four essential technical designs of 20 widely adopted optimizers, as described in Algorithm 1. We classify these optimizers based on their strategies of the learning rate adjustment (step 2) and the gradient estimation (step 3). Specially, we consider five types of statistics during training: (i) *First(-order) moment (gradient)*: The gradient itself, the first-order partial derivative of the objective function concerning the parameters. (ii) *Estimated first-moment gradient (momentum)*: An exponentially decaying average of past gradients, serving as an estimate of the first-order moment. (iii) *Second(-order) moment (gradient)*: The second-order partial derivative of the objective function for the parameters, also known as the Hessian matrix, which can be approximated by Nesterov gradient descenting (Reddi et al., 2018; Xie et al., 2023). (iv) *Estimated second moment*: An exponential moving average (EMA) of the squared gradients, providing an estimate of the second-order moment. (v) *Second-order gradient*: The Hessian matrix, the second-order partial derivative of the objective function to the parameters.

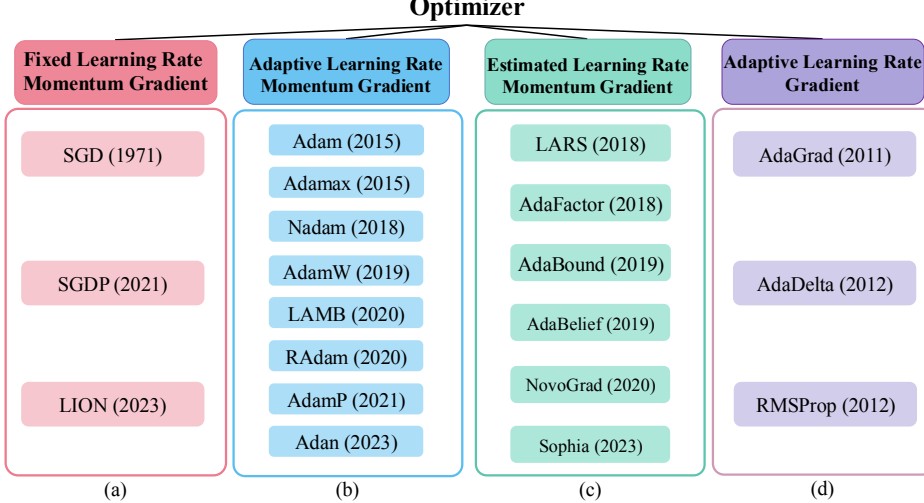

Figure A1: Overview of mainstream gradient-based optimizers, which are categorized by the techniques of learning rate adjustment (step 2) and gradient estimation (step 3) in Algorithm 1. (a) and (d) only optionally employs step 2 (momentum gradients) or step 3 (adaptive learning rates), while (b) and (c) consider both of them. (b) employs adaptive learning rates by estimating second moments; (c) estimates the dynamic learning rate by other gradient components except for the second moments.

Table A2: Four categories of typical optimizers with their components. From top to bottom are (a) fixed learning rate with momentum gradient, (b) adaptive learning rate with momentum gradient, (c) estimated learning rate with momentum gradient, and (d) adaptive learning rate with current gradient.

| Optimizer | Date | Learning rate | Gradient | Weight decay |
|---|---|---|---|---|
| SGD-M (Sinha & Griscik, 1971) | TSMC'1971 | Fixed lr | Momentum | ✓ |
| SGDP (Heo et al., 2021) | ICLR'2021 | Fixed lr | Momentum | Decoupled |
| LION (Chen et al., 2023) | NIPS'2023 | Fixed lr | Sign Momentum | Decoupled |
| Adam (Kingma & Ba, 2015) | ICLR'2015 | Estimated second moment | Momentum | ✓ |
| Adamax (Kingma & Ba, 2015) | ICLR'2015 | Estimated second moment | Momentum | ✓ |
| AdamW (Loshchilov & Hutter, 2019) | ICLR'2019 | Estimated second moment | Momentum | Decoupled |
| AdamP (Heo et al., 2021) | ICLR'2021 | Estimated second moment | Momentum | Decoupled |
| LAMB (You et al., 2020) | ICLR'2020 | Estimated second moment | Momentum | Decoupled |
| NAdam (Reddi et al., 2018) | ICLR'2018 | Estimated second moment | Nesterov Momentum | ✓ |
| RAdam (Liu et al., 2020) | ICLR'2020 | Estimated second moment | Momentum | Decoupled |
| Adan (Xie et al., 2023) | TPAMI'2023 | Estimated second moment Nesterov | Momentum | Decoupled |
| AdaBelief (Zhuang et al., 2020) | NIPS'2019 | Estimated second moment variance | Momentum | Decoupled |
| AdaBound (Luo et al., 2019) | ICLR'2019 | Estimated second moment | Momentum | Decoupled |
| AdaFactor (Shazeer & Stern, 2018) | ICML'2018 | Estimated second moment (decomposition) | Momentum | Decoupled |
| LARS (Ginsburg et al., 2018) | ICLR'2018 | L2-norm of Gradient | Momentum | Decoupled |
| Novograd (Ginsburg et al., 2020) | arXiv'2020 | Sum of estimated second momentum | Momentum | Decoupled |
| Sophia (Liu et al., 2023) | arXiv'2023 | Parameter-based estimator | Sign Momentum | Decoupled |
| AdaGrad (Duchi et al., 2011) | JMLR'2011 | Second moment | Gradient | ✓ |
| AdaDelta (Zeiler, 2012) | arXiv'2012 | Estimated second moment param moment | Gradient | ✓ |
| RMSProp (Hinton, 2012) | arXiv'2012 | Estimated second moment | Gradient | ✓ |

# B   IMPLEMENTATION DETAILS

This section provides experimental settings of benchmarks and dataset information for Sec 3. We benchmarked 16 typical vision networks as discussed in Sec. 2.1 with the image classification task with the following benchmark settings. We apply consistent setups for image classification tasks on CIFAR-100 (Krizhevsky et al., 2009) and ImageNet-1K (Krizhevsky et al., 2012b) based on OpenMixup (Li et al., 2022) codebase with 1 or 8 Nvidia A100 GPUs. As for transfer learning with pre-trained models, we employ object detection and pose estimation tasks (Ren et al., 2015) on COCO (Lin et al., 2014) with MMLab codebases (Chen et al., 2019).

## B.1   IMAGE CLASSIFICATION

**ImageNet-1K.** Following the widely used modern training recipes, we consider three regular training settings for ImageNet-1K (Krizhevsky et al., 2012b) classification experiments for various backbones and optimizers, which could be transplanted to the proposed CIFAR-100 benchmarks. As shown in Table A3, these training schemes include data preprocessing and augmentations, optimizing setups, regularization tricks, and loss functions: (1) Classical **PyTorch-style setting** (Szegedy

et al., 2016) applies basic data augmentations, `RandomResizeCrop` (or `RandomCrop` for $32^2$ resolutions), `HorizontalFlip`, and `CenterCrop` (Szegedy et al., 2015), basic SGD training setups with cosine learning rate scheduler (Loshchilov & Hutter, 2016), and the cross-entropy (CE) loss. (2) **DeiT and ConvNeXt settings** (Touvron et al., 2021a; Liu et al., 2021) are designed for Transformer and modern CNN architectures like ViTs (Dosovitskiy et al., 2021; Graham et al., 2021), which utilizes several advanced augmentations (Cubuk et al., 2019) (like RandAugment (Cubuk et al., 2020), Mixup (Zhang et al., 2018) and CutMix (Yun et al., 2019; Liu et al., 2022b), Random Erasing (Zhong et al., 2020), ColorJitter (He et al., 2016)), and regularization techniques (Stochastic Depth (Huang et al., 2016), Label Smoothing (Szegedy et al., 2016), and EMA (Polyak & Juditsky, 1992). (3) **RSB A2/A3 settings** (Wightman et al., 2021) are designed for CNNs to boost their performance and convergence speeds as ViTs, which reduces the augmentation strengths and replaces the CE loss with Binary Cross Entropy (BCE) loss compared to the DeiT setting. The optimizing hyper-parameters marked in gray, like initial learning rate, optimizer momentum, and weight decay, will be tuned based on the optimizer. We use the threshold $\lambda = 1$ in Eq. (1) to discriminate BOCB results on ImageNet-1K.

Table A3: Ingredients used for image classification training settings. Taking ImageNet-1K as the template setup, the settings of PyTorch (Szegedy et al., 2016) and RSB A2/A3 (Wightman et al., 2021) take ResNet-50 (He et al., 2016) for instances, the DeiT (Touvron et al., 2021a) setting takes DeiT-S as the example, and the ConvNeXt (Liu et al., 2022a) setting is a variant of the DeiT setting for ConvNeXt and Swin Transformer (Liu et al., 2021). Gray regions will be tuned for each optimizer.

| Procedure | PyTorch | | DeiT | | ConvNeXt | RSB A2 | | RSB A3 |
|---|---|---|---|---|---|---|---|---|
| Dataset | IN-1K | CIFAR | IN-1K | CIFAR | CIFAR | IN-1K | CIFAR | IN-1K |
| Train Resolution | 224 | 224 | 224 | 224 | 32 | 224 | 224 | 160 |
| Test Resolution | 224 | 224 | 224 | 224 | 32 | 224 | 224 | 224 |
| Test crop ratio | 0.875 | 1.0 | 0.875 | 1.0 | 1.0 | 0.95 | 1.0 | 0.95 |
| Epochs | 100 | 200 | 300 | 200 | 200 | 300 | 200 | 100 |
| Batch size | 256 | 100 | 1024 | 100 | 100 | 2048 | 100 | 2048 |
| Optimizer | SGD | | AdamW | | AdamW | LAMB | | LAMB |
| Learning rate | 0.1 | | $1 \times 10^{-3}$ | | $1 \times 10^{-3}$ | $5 \times 10^{-3}$ | | $8 \times 10^{-3}$ |
| Optimizer Momentum | 0.9 | | 0.9, 0.999 | | 0.9, 0.999 | 0.9, 0.999 | | 0.9, 0.999 |
| Weight decay | $10^{-4}$ | | 0.05 | | 0.05 | 0.02 | | 0.02 |
| LR decay | Cosine | | Cosine | | Cosine | Cosine | | Cosine |
| Warmup epochs | ✗ | | 5 | | 20 | 5 | | 5 |
| Label smoothing $\epsilon$ | ✗ | | 0.1 | | 0.1 | ✗ | | ✗ |
| Dropout | ✗ | | ✗ | | ✗ | ✗ | | ✗ |
| Stochastic Depth | ✗ | | 0.1 | | 0.1 | 0.05 | | ✗ |
| Repeated Augmentation | ✗ | | ✓ | | ✓ | ✓ | | ✗ |
| Gradient Clip. | ✗ | | 5.0 | | ✗ | ✗ | | ✗ |
| Horizontal flip | ✓ | | ✓ | | ✓ | ✓ | | ✓ |
| RandomResizedCrop | ✓ | | ✓ | | ✓ | ✓ | | ✓ |
| Rand Augment | ✗ | | 9/0.5 | | 9/0.5 | 7/0.5 | | 6/0.5 |
| Auto Augment | ✗ | | ✗ | | ✗ | ✗ | | ✗ |
| Mixup $\alpha$ | ✗ | | 0.8 | | 0.8 | 0.1 | | 0.1 |
| Cutmix $\alpha$ | ✗ | | 1.0 | | 1.0 | 1.0 | | 1.0 |
| Erasing probability | ✗ | | 0.25 | | 0.25 | ✗ | | ✗ |
| ColorJitter | ✗ | | ✗ | | ✗ | ✗ | | ✗ |
| EMA | ✗ | | ✓ | | ✓ | ✗ | | ✗ |
| CE loss | ✓ | | ✓ | | ✓ | ✗ | | ✗ |
| BCE loss | ✗ | | ✗ | | ✗ | ✓ | | ✓ |

**CIFAR-100.** Inheriting the training settings on ImageNet-1K, we modify the input resolutions and batch size to build the corresponding settings for CIFAR-100 (Krizhevsky et al., 2009) benchmarks. The original CIFAR-100 dataset contains 50k training images and 10k testing images in $32^2$ resolutions, and we consider two input resolutions. As shown in Table A3, in the case of $32^2$ resolutions, the downsampling ratio of the first stem in CNNs will be set to $\frac{1}{2}$; in the case of $224^2$ resolutions (cubic upsampling to $224^2$), the backbone structure keep the same as on ImageNet-1K. We use different training settings for a fair comparison of classical CNNs and modern Transformers on CIFAR-100, which contains 50k training images and 10k testing images of $32^2$ resolutions. As for classical CNNs with bottleneck structures, we use $32^2$ resolutions with the CIFAR version of network architectures, *i.e.,* downsampling the input size to $\frac{1}{2}$ in the stem module instead of $\frac{1}{8}$ on ImageNet-1K. All the benchmarked backbones are trained for 200 epochs from the stretch. We set $\lambda = 3$ in Eq. (1) to discriminate BOCB results on CIFAR-100.

**Optimizing hyper-parameters search.** For a fair comparison, we only search two common hyper-parameters (the learning rate and weight decay) heuristically with NNI toolbox (Microsoft, 2021), *i.e.,* determining the NNI search range of hyper-parameters manually. We regard each hyper-parameter as a set of discrete values, choosing 5 consecutive values centered on the heuristically determined initial value. As for the specific hyper-parameters of some optimizers, *e.g.*, $\epsilon$ for AdaBelief and the final lr for AdaBound, we further search for their optimal values separately. Table A1 shows the training setting for each backbone The basis hyper-parameters of various optimizers for different vision backbones on CIFAR-100 are provided in the supplementary material.

### B.2 OBJECT DETECTION AND POSE ESTIMATION

**Object Detection.** Following Swin Transformers (Liu et al., 2021), we first evaluate objection detection as the representative vision downstream task on COCO (Lin et al., 2014) for transfer learning, which includes 118K training images (*train2017*) and 5K validation images (*val2017*). Experiments of COCO detection and segmentations are implemented on MMDetection (Chen et al., 2019) codebase and run on 4 Tesla V100 GPUs. Taking RetinaNet (Lin et al., 2017) as the standard detector, the original fine-tuning setting for ResNet-50 employs the SGD optimizer with $1\times$ (12 epochs) training with a batch size of 16 and a fixed step learning rate scheduler. As for Swin-T, the official setting employs the AdamW optimizer with $1\times$ scheduler and a batch size of 16. During training, the shorter side of training images is resized to 800 pixels, and the longer side is resized to not more than 1333 pixels. For different pre-trained models (PyTorch, DeiT, and RSB A2/A3 pre-training), we search basic hyper-parameters (the learning rate and the weight decay) for every optimizer as described in Sec. B.1 to get relatively optimal results. We set $\lambda = 3$ in Eq. (1) to discriminate BOCB results for objection detection.

**2D Pose Estimation.** We also evaluate transfer learning to 2D human key-points estimation task on COCO based on Top-Down SimpleBaseline (Xiao et al., 2018) (adding a Top-Down estimation head after the backbone) following MogaNet (Li et al., 2024). The original training setting is to fine-tune the pre-trained backbone and the randomly initialized head for 210 epochs with Adam optimizer with a multi-step learning rate scheduler decay at 170 and 200 epochs. We also search learning rates and weight decays for all optimizers. The training and testing images are resized to $256 \times 192$ or $384 \times 288$ resolutions, and these experiments are implemented with MMPose (Contributors, 2020) codebase and run on 4 Tesla V100 GPUs. We set $\lambda = 3$ in Eq. (1) to discriminate BOCB results for the pose estimation task.

### B.3 EMPRICIAL ANALYSIS

To gain deeper insights into the observed *backbone-optimizer coupling bias* (BOCB) phenomenon, we conducted a collection of empirical analysis focusing on two key aspects: hyper-parameter stability and model parameter patterns. These analyses provide valuable information about the intrinsic properties of different network architectures and their interactions with various optimizers.

**Hyper-parameter stability.** We developed an approach to quantify the hyper-parameter stability of vision backbones and optimizers, which serves as a proxy for understanding the strength of backbone-optimizer coupling. This analysis involves the following steps: (1) Optimal Settings Identification: For each backbone-optimizer pair, we conducted extensive grid searches to identify the optimal hyper-parameters (learning rate and weight decay). (2) One-hot Encoding: We converted these optimal hyper-parameters into discrete one-hot encoded vectors. Assuming $n$ possible learning rates and $m$ possible weight decays, we created vectors $\{\tilde{lr}_i\}_{i=1}^{n}$ and $\{\tilde{\omega}_i\}_{i=1}^{m}$. (3) Mode Statistics: We computed histogram-based mode (most common) statistics $M_{lr}$ and $M_{\omega}$ across all optimizers for each backbone. (4) Variation Computation: We quantified the *variation* between each hyper-parameter and mode statistics using the Manhattan distance, $\Sigma_{i=1}^{n}|\tilde{lr}_i - M_{lr}| + \Sigma_{i=1}^{m}|\tilde{\omega}_i - M_{\omega}|$. (5) Visualization: We plot the distribution of these variations for both backbones (Figure 4) and optimizers (Figure 5), which offer intuitive insights into the relative stability and adaptability of different backbone-optimizer pairs. As for backbones, lower variation indicates higher stability and potentially weaker coupling bias, as the backbone performs well across a range of optimizers with similar hyper-parameters. For the optimizers, lower variation suggests better generalizability across different network architectures.

**Patterns of learned parameters.** To investigate the layer-wise properties discussed in Section 2.1, we employed a set of quantitative metrics to analyze the learned parameters of each layer. As shown in

Section D, these metrics reveal intrinsic topological patterns that reflect the unique characteristics of different network architectures, such as stage-wise macro designs, building block structures, and core operators of token-mixers and channel-mixers. We focused on the three key indicators as follows:

- **PL Exponent Alpha**: In the context of `WeightWatcher` (Martin & Mahoney, 2021; Martin et al., 2021), the Power Law (PL) exponent $\alpha$ quantifies the learned parameter quality of neural network layers. It is extracted from the tail fit of the layer weight matrix's Empirical Spectral Density (ESD) to a truncated Power Law: $\rho(\lambda) \sim \lambda^{-\alpha}$, $\rho(\lambda)$ denotes the ESD, and $\lambda$ represents the eigenvalues of the weight matrix's correlation matrix $X = W^T W$. The exponent $\alpha$ reflects the correlation structure, with lower values indicating enhanced generalization capabilities and higher values suggesting potential overfitting or underfitting. This metric facilitates the assessment of neural network models' generalization tendencies without the need for training or testing datasets, serving as an intrinsic measure of model quality.

- **Entropy**: The information entropy of the learned parameter tensor, $H = -\sum p_i \log(p_i)$, where $p_i$ is the probability of each value in the parameter tensor. It is used to measure the randomness of the parameter distribution. Higher entropy indicates a more uniform or random distribution, while lower entropy suggests a more patterned distribution. This provides insights into the complexity and information of each layer, helping to identify layers with more structured weight distributions.

- $L_2$**-norm**: Euclidean norm (magnitude) of the learned parameter vector $||w||_2 = \text{sqrt}(\sum w_i^2)$, where $w_i$ are individual parameters. This reflects the scale of the learned weight matrix and identifies layers with potential dominant effects on the network's behavior (more influence on the layer output), which could be crucial for understanding the learning results of diverse network architectures.

- **Top-$k$ PCA Energy Ratio**: Cumulative energy ratio of the top-$k$ principal components of the parameter matrix $E_k = (\sum_{i=1}^{k} \lambda_i)/(\sum_{i=1}^{n} \lambda_i)$, where $\lambda_i$ are eigenvalues of the covariance matrix. It measures the concentration of information in the learned parameter matrix. A larger top-$k$ energy indicates that the parameter matrix has more concentrated components. This analysis provides insights into the dimensionality and compressibility of each layer's parameters, which could be helpful for model pruning and efficiency optimization.

These metrics, when analyzed across different layers and backbone-optimizer combinations, reveal characteristic patterns that correspond to specific architecture designs. We provide ridge plots (as shown in Section D) to visualize these metrics across different layers for various backbone-optimizer combinations. For instance, we may observe distinct entropy patterns in hierarchical vs. isotropic stage-wise architectures, variations in $L_2$-norm across different stages of the network, or changes in PCA energy ratios for different types of layers (e.g., convolutional vs. attention-based).

By analyzing these patterns, we can gain valuable insights into how different neural network architectures interact with various optimizers, furthering our understanding of the BOCB phenomenon and informing future design choices for both vision backbones and optimizers.

## C  FULL EXPERIMENTAL RESULTS

This appendix section provides a detailed expansion of the experimental findings from the main manuscript, specifically aimed at validating the BOCB phenomenon. The results are structured to facilitate a thorough evaluation across the CIFAR-100 and ImageNet-1K datasets, involving a diverse range of both modern and classical vision backbones, each paired with various optimizers. This comprehensive analysis is intended to clarify the complex interactions between neural network architectures and optimization strategies, emphasizing their critical impact on model performance and adaptability. Additionally, these insights are applied to practical tasks, such as object detection and pose estimation on COCO, demonstrating the practical relevance of BOCB.

### C.1  CIFAR-100 CLASSIFICATION EXPERIMENTS

Our in-depth exploration of the CIFAR-100 dataset was designed to scrutinize the interdependence between network architectures and optimizers. Table 1 encapsulates the top-1 classification accuracy for an extensive lineup of 15 vision backbones, categorized into primary CNNs, classical CNNs, and modern DNNs. The results underscore a pronounced divergence in the optimal optimizer for different architectural eras. Classic architectures such as AlexNet, VGG, and the ResNet family reveal an affinity for SGD-M and SGDP, with these optimizers yielding the most accurate outcomes. This preference indicates a tight coupling between classical CNNs and SGD-based methods. In stark contrast,

modern architectures like Vision Transformers, ConvNeXt, and MetaFormer variants thrive under the adaptive learning rates afforded by optimizers such as AdamW, AdamP, and LAMB, showcasing a more flexible coupling bias. To elucidate the nuances of BOCB, we present a hyperparameter sensitivity analysis. This analysis visualizes the distribution of optimal learning rates and weight decays for the evaluated optimizers, as depicted in Figures 3 and 4. Classical CNNs display a concentrated distribution, pointing to a specific hyperparameter set for SGD optimizers. In contrast, modern DNNs exhibit a broader distribution, suggesting a higher tolerance to hyperparameter variations and a more adaptable coupling with a range of optimizers.

## C.2  IMAGENET-1K CLASSIFICATION EXPERIMENTS

To ascertain the generalizability of our observations, we extended our evaluation on ImageNet-1K. Table 2 details the Top-1 accuracy for a curated selection of vision backbones under various optimizers. The results are congruent with those from CIFAR-100, reinforcing the BOCB phenomenon. ResNets and Efficient-Nets continue demonstrating their predilection for SGD-M and SGDP, achieving peak performance with these optimizers. On the other hand, modern DNNs like Vision Transformers and ConvNeXt once again manifest their superiority when paired with AdamW, AdamP, and LAMB, aligning with the adaptive learning rate optimiz-

Table A4: Top-1 accuracy (%) of various vision backbones training 300 epochs by three optimal optimizers and five indicator optimizers with DeiT or RSB A2 settings on ImageNet-1K.

| Optimizer | R-50 | DeiT-S | CNX-T | CNXV2-T | CF-12 |
|---|---|---|---|---|---|
| AdamW | 79.9 | 80.4 | 82.1 | 82.3 | 81.6 |
| LAMB | 79.8 | 80.2 | 82.2 | 82.3 | 81.5 |
| Adan | 79.9 | 80.8 | 82.6 | 82.8 | 81.8 |
| SGD | 78.8 | 75.4 | 71.3 | 76.8 | 79.7 |
| AdaBound | 75.4 | 73.0 | 72.4 | 77.1 | 79.6 |
| LARS | 79.7 | 73.2 | 75.9 | 79.6 | 79.9 |
| RMSProp | 78.0 | 78.0 | 79.6 | 80.2 | 80.4 |
| AdaDelta | 74.9 | 55.0 | 73.5 | 77.9 | 78.5 |
| Std/Range | 1.9/5.0 | 7.9/25.8 | 4.4/11.3 | 2.3/6.0 | 1.1/3.3 |

ers' capacity to navigate the complex optimization landscapes of contemporary architectures. We also verify our findings in Sec. 4.2, as shown in Table A4, ResNet-50 and ConvFormer-S12 show weak BOCB properties while DeiT-S and ConvNeXt-T have strong coupling bias with AdamW-like optimizers. ConvNeXt.V2 improves the performance and BOCB property of ConvNeXt with the certain design GRN (Woo et al., 2023) between the FFN modules.

## C.3  COCO OBJECT DETECTION AND POSE ESTIMATION EXPERIMENTS

Expanding our analysis from CIFAR-100 and ImageNet-1K, we investigated the BOCB in practical tasks using COCO for object detection and pose estimation. These experiments aimed to assess BOCB's impact on model transferability and task performance when pre-trained models are adapted to specific tasks. In object detection, employing the RetinaNet framework with ImageNet-1K pre-trained models, we observed in Table 3 that backbones trained with adaptive optimizers like AdamW, AdamP, and LAMB achieved higher top-1 accuracies on ImageNet-1K and superior performance on COCO object detection. This suggests that these optimizers enhance feature learning and generalization in downstream tasks by effectively navigating complex optimization landscapes during pre-training.

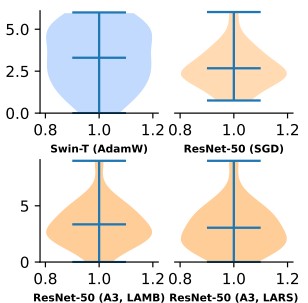

Figure A2: **Violinplot** of hyperparameters for the aspects of backbones or optimizers on COCO.

Similarly, for pose estimation using the TopDown approach, models pre-trained with AdamW, AdamP, and LAMB showed improved performance on COCO, as evidenced by higher AP50 scores in Table 3. This supports the significant influence of the optimizer choice during pre-training on a model's capacity to acquire and transfer knowledge. Our hyperparameter sensitivity analysis, extended to COCO experiments, provides further insights. Figure A2 illustrates the distribution of optimal learning rates and weight decays for various optimizers, revealing that while classical backbones have a narrow optimal range, modern architectures display broader tolerance, reflecting their adaptability to different optimizer settings. This adaptability is crucial for effective transfer learning and task-specific performance.

In summary, the comprehensive experimental results presented in this section provide compelling evidence for the backbone-optimizer coupling bias phenomenon across multiple benchmark datasets and vision tasks. These findings highlight the importance of considering the interplay between network architectures and optimization algorithms when designing and deploying vision systems, as overlooking BOCB can lead to suboptimal performance and potential inefficiencies.

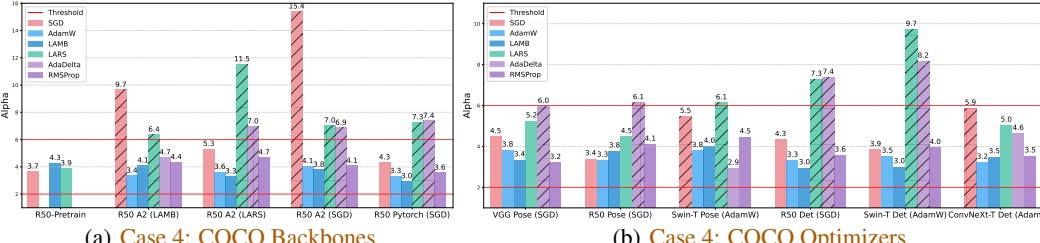

(a) Case 4: COCO Backbones      (b) Case 4: COCO Optimizers

Figure A3: **BOCB case studies: PL exponent alpha metrics of different backbones and pre-training optimizers** on COCO. The alpha metric (Martin et al., 2021) measures the fitting quality of models to a certain task, and a smaller alpha value indicates better fitting. These diagonal bars denote the BOCB occurring. Please refer to the details on the alpha metric in Appendix B.3.

## D EMPRICIAL EXPERIMENTS

This section delineates a series of empirical experiments meticulously designed to validate the theoretical insights into the BOCB and to elucidate the nuances of this phenomenon within the context of network architecture and optimization strategies. The experiments are crafted to furnish a comprehensive understanding of BOCB, its implications for vision backbones, and its interaction with various optimization techniques.

### D.1 MACRO DESIGN'S INFLUENCE ON OPTIMIZATION

Our empirical inquiry commenced with a profound analysis of the macro design's impact on the optimization landscape. We executed extensive experiments utilizing a diverse array of vision backbones, ranging from Primary CNNs, which laid the groundwork for the CNN paradigm, through classical CNNs such as ResNet, which introduced a stage-wise hierarchical design, to Modern DNNs like ConvNeXt and MogaNet, which feature complex block-wise heterogeneous structures.

Our findings, as depicted in Figure 1, unveil a discernible trend: the escalation of macro design complexity corresponds with an increase in optimization complexity. This is notably evident in the juxtaposition between ResNet-50 and contemporary backbones such as MobileNetV2 and Efficient-Net. While ResNet-50, with its stage-wise hierarchical architecture, exhibits a robust coupling with SGD optimizers, the latter backbones manifest a predilection for adaptive learning rate optimizers due to their intricate feature extraction mechanisms.

### D.2 TOKEN MIXING AND OPTIMIZATION SYNERGIES

In our quest to unravel the effects of token-mixing operations on optimization, we scrutinized the performance of various token-mixing operators within the MetaFormer architecture. As meticulously detailed in Table 1, each token mixing operator—Identity, Pooling, Attention, and Convolution—presents unique challenges and sensitivities to optimizer hyperparameters.

The ConvFormer architecture, as a MetaFormer derivative, epitomizes a balanced approach to token mixing and optimization. By adopting a streamlined block-wise design and alternating between convolutional and token mixing blocks, ConvFormer mitigates BOCB and facilitates a more efficient optimization process. This approach underscores the significance of harmonizing architectural design with optimization strategies to minimize BOCB.

### D.3 OPTIMIZER SELECTION AND THE BOCB NEXUS

To gauge the impact of optimizer selection on BOCB, we conducted experiments with a panoply of optimizers across diverse backbones. The results, as illustrated in Figure 5, indicate that the choice of optimizer significantly modulates the extent of BOCB. Optimizers adept at navigating complex optimization landscapes, such as those in Categories (b) and (c), exhibit robust performance across a spectrum of backbones. Conversely, Category (a) optimizers necessitate meticulous hyperparameter tuning for classical CNNs, while Category (d) optimizers manifest the most pronounced BOCB and suboptimal performance.

Our empirical analysis accentuates the critical interplay between network macro design, token mixing operations, and optimizer selection in sculpting the optimization landscape of vision backbones. The

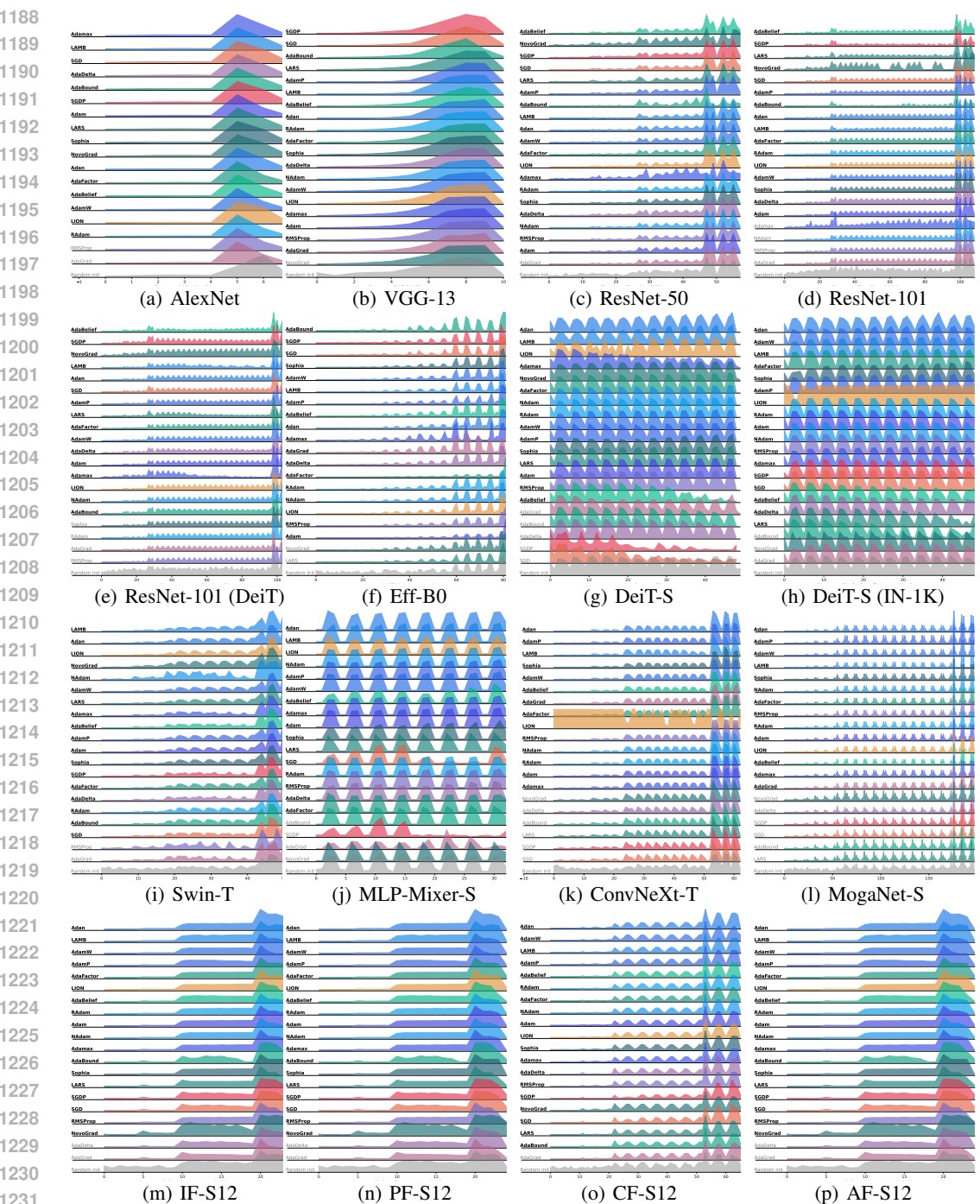

Figure A4: Ridge plot of the entropy of learned parameters on CIFAR-100. For the sub-figure of each optimizer, the X and Y axes indicate the layer indexes and the entropy of weights.

findings offer valuable insights for designing future vision backbones, emphasizing the imperative for a balanced approach that aligns backbone design with selecting appropriate optimizers.

## D.4 PRE-TRAINING AND TRANSFER LEARNING

Extending our investigation to practical applications, we examined the performance of various optimizers in the context of pre-training on ImageNet-1K and subsequent transfer learning to tasks such as object detection with RetinaNet and pose estimation on COCO. As demonstrated in Table 1,

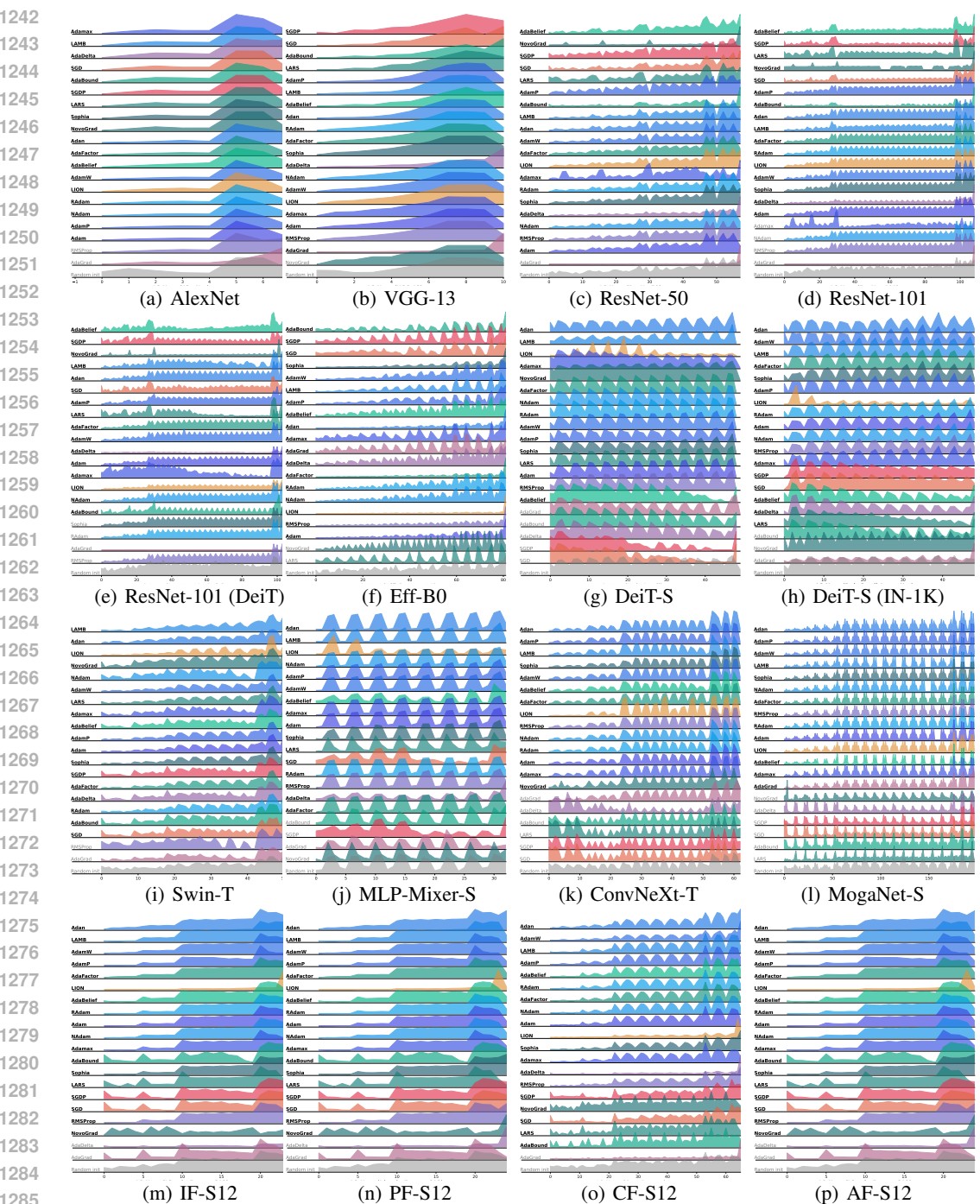

Figure A5: Ridge plot of the $L_2$-norm of learned parameters on CIFAR-100. For the sub-figure of each optimizer, the X and Y axes indicate the layer indexes and the $L_2$-norm of weights.

optimizers like AdamW, which exhibited a reliable peak in performance during pre-training, sustained their superiority in transfer learning scenarios. This suggests that the choice of optimizer during the pre-training phase can significantly influence the transfer learning outcomes.

Our experiments also underscore the importance of a comprehensive pre-training phase that pairs vision backbones with suitable optimizers to ensure robust transfer learning capabilities. Models that underwent an extended pre-training period with optimizers like LAMB demonstrated enhanced performance compared to those with shorter pre-training durations using SGD or other optimizers.

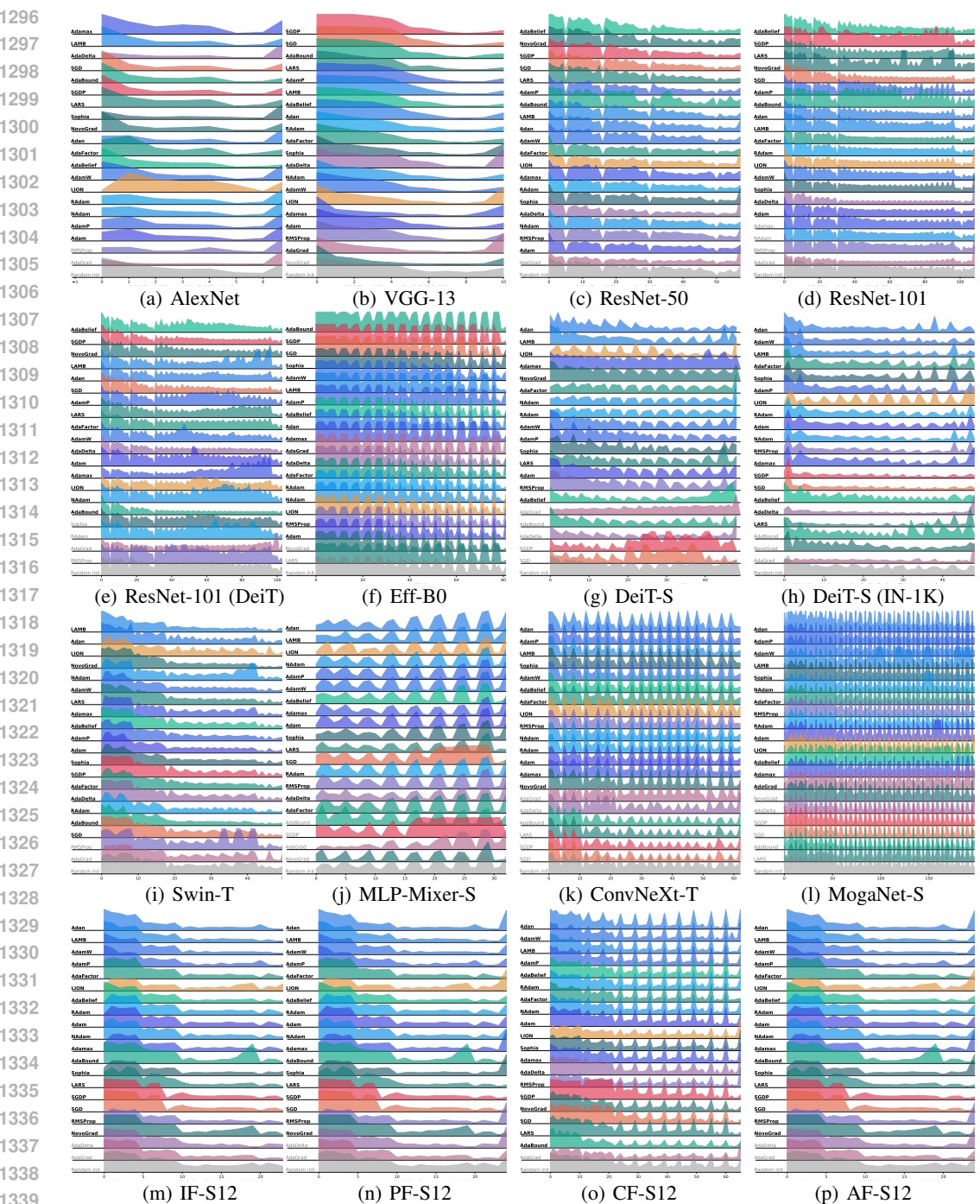

Figure A6: Ridge plot of the top-K energy PCA ratio of learned parameters on CIFAR-100. For the sub-figure of each optimizer, the X and Y axes indicate the layer indexes and the top-K PCA ratio of weights. Weights with a larger top-k PCA ratio yield skewed eigenvalue distributions, making these plots show opposite values as plots with entropy or $L2$-norm.

The empirical experiments presented in this section provide a robust validation of the BOCB phenomenon and its implications for the design and optimization of vision backbones. By systematically exploring the interplay between network macro design, token mixing operations, and optimizer selection, we have identified key factors that contribute to BOCB and provided actionable guidelines for mitigating its impact. Our findings underscore the need for a balanced approach to backbone

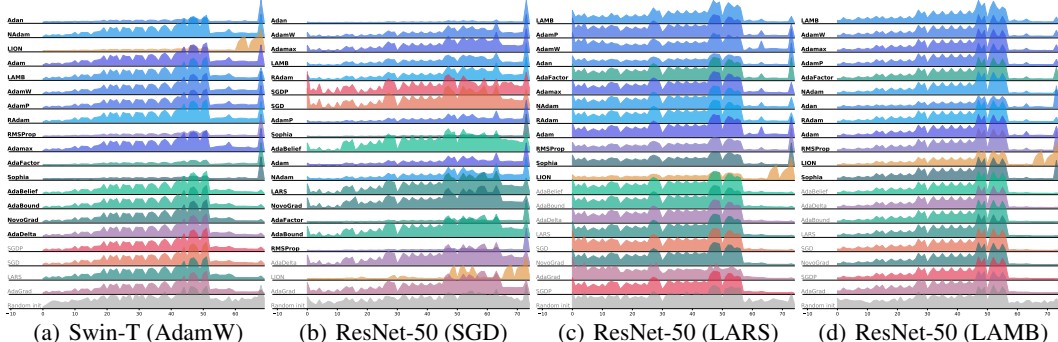

|  (a) Swin-T (AdamW) | (b) ResNet-50 (SGD) | (c) ResNet-50 (LARS) | (d) ResNet-50 (LAMB) |

Figure A7: Ridge plot of the $L_2$-norm parameter patterns for transfer learning to object detection (RetinaNet) based on Swin-T and ResNet-50 on COCO, where (a)-(d) are pre-trained by AdamW, SGD, LARS, and LAMB optimizers on ImageNet-1K. Notably, the distributions of backbone parameters are largely determined by pre-training, while the randomly initialized weights of FPN and detection head (after the 53-th or 58-th layer in Swin-T and ResNet-50) distinguish the trial patterns.

design and optimizer selection to enhance training efficiency and performance in computer vision applications.

Table A5: Rankings of optimizers with various aspects for practical usage. Benchmarked results rank the properties of performance and hyper-parameter robustness, the BOCB property is marked as 1 or 0, and the computational overhead is ranked by the average training hours. As described in Appendix D.5, the overall ranking is estimated as the task-home message for selecting optimizers.

|  | SGD | SGDP | LION | Adam | Adamax | NAdam | AdamW | LAMB | RAdam | AdamP | Adan | AdaBound | LARS | AdaFactor | AdaBelief | NovoGrad | Sophia | AdaGrad | AdaDelta | RMSProp |
|---|---|---|---|---|---|---|---|---|---|---|---|---|---|---|---|---|---|---|---|---|
| Performance | 17 | 15 | 11 | 9 | 12 | 8 | 5 | 2 | 10 | 3 | 1 | 14 | 19 | 6 | 4 | 13 | 7 | 20 | 16 | 18 |
| Hyper-parameter | 10 | 6 | 3 | 2 | 7 | 4 | 1 | 1 | 8 | 5 | 9 | 15 | 14 | 6 | 6 | 13 | 7 | 11 | 12 | 1 |
| BOCB | 1 | 1 | 1 | 0 | 1 | 1 | 0 | 0 | 0 | 1 | 0 | 1 | 1 | 0 | 1 | 1 | 0 | 1 | 1 | 1 |
| Computations | 1 | 2 | 2 | 5 | 5 | 4 | 5 | 5 | 5 | 5 | 6 | 5 | 2 | 2 | 5 | 2 | 5 | 2 | 5 | 2 |
| Overall | 16 | 13 | 10 | 7 | 12 | 8 | 2 | 1 | 11 | 4 | 3 | 17 | 20 | 6 | 5 | 15 | 9 | 19 | 18 | 14 |

## D.5 RULES FOR COUNTING THE OPTIMIZER RANKINGS

We have summarized and analyzed a great number of mixup benchmarking results to compare and rank all the included mixup methods in terms of *performance*, *applicability*, and the *overall* capacity. We have conducted a comprehensive meta-analysis of optimizer benchmarking results to systematically evaluate and rank a diverse array of optimization algorithms across four critical dimensions: *Performance*, *Hyperparameter Robustness*, *Backbone Optimizer Coupling Bias* (BOCB), and *Computational Efficiency*. Our methodology employs a weighted scoring system to synthesize these multifaceted evaluations:

- **Performance** (40% weight): This metric quantifies an optimizer's efficacy across various backbone architectures, reflecting its paramount importance in algorithm selection.

- **Hyperparameter Robustness** (20% weight): Quantified as the median Manhattan distance from the optimal learning rate and weight decay configurations to the maximum average distance, this metric assesses the optimizers' robustness to hyperparameter perturbations.

- **BOCB** (20% weight): Represented as a binary indicator (1 or 0), this factor evaluates the potential for coupling deviation between the optimizer and the backbone architecture.

- **Computational Efficiency** (20% weight): Measured by GPU memory allocation, this dimension quantifies the computational resources required by each optimizer.

The aggregation of these standardized scores yields a comprehensive ranking that serves as a robust benchmark for optimizer selection in deep learning visual backbone scenarios. This multidimensional analysis not only elucidates the relative merits of established algorithms such as AdamW—corroborating its long-standing prevalence in the community—but also highlights the potential of emerging optimizers like Adan and LAMB, particularly in contexts where BOCB or

hyperparameter robustness are of paramount importance. Meanwhile, we also recognized some optimizers could be sensitive and served as the indicator to show whether the given backbone has the potential of BOCB. Hence, we summarize two groups of optimizers as follows:

- **High-performance Optimizers**: AdamW (Loshchilov & Hutter, 2019), LAMB (You et al., 2020), and Adan (Xie et al., 2023) help the most networks perform well in various scenarios.
- **BOCB Indicator Optimizers**: Conducting benchmarks with SGD (Sinha & Griscik, 1971), AdaBound (Luo et al., 2019), LARS (Ginsburg et al., 2018), RMSProp (Hinton, 2012), and AdaDelta (Zeiler, 2012) could help users recognize whether a given backbone architecture has the risk of BOCB on a new scenario.

## E    LIMITATIONS

This work has several limitations: (1) Although we conduct transfer learning experiments to ImageNet and COCO, the benchmark is mainly focused on CIFAR-100, which may lead to questionable findings for broader downstream tasks. However, all our transfer learning results are consistent with the CIFAR-100 findings. Moreover, our released code can be easily extended to other tasks. Users can thus easily conduct validations with it. (2) BOCB is more complex than current metrics such as parameters and FLOPs, which may lead to inconvenience in practice. We suggest researchers use our code, selecting representative optimizers, such as SGD, Adam, and AdamW, for the ridge plot validation and CIFAR-100 benchmarks, which are practical and resource-efficient. We also call for further explorations of BOCB in the community to advance vision systems together.

