# OpenReview forum: "Unveiling the Backbone-Optimizer Coupling Bias in Visual Representation Learning"
_ICLR.cc/2025/Conference — Submitted to ICLR 2025_

### Official Review · Reviewer_FcXi · 2024-10-23

**Soundness:** 2
**Presentation:** 3
**Contribution:** 2
**Rating:** 6
**Confidence:** 3

**Summary:**

This paper introduces the concept of Backbone-Optimizer Coupling Bias (BOCB) -- i.e., strong dependencies between neural network backbones and specific optimizers -- which affects performance and generalization. To demonstrate this, a comprehensive benchmark is provided, evaluating various vision backbones and popular optimizers across different tasks.

**Strengths:**

* I believe the problem statement of the paper is promising. Even though it is known that some modern architectures need specific optimizers to achieve high performance (e.g., ViTs use AdamW by default these days), there are only a few comprehensive studies on the relationship between architectures and optimizers.
* This paper provides results using a wide range of optimizers and architectures, and it reports the average performance over three runs.
* This paper not only reports the phenomenon but also tries to reveal the underlying causes.

**Weaknesses:**

* One of the major weaknesses is soundness. I am not fully convinced by some statements in the introduction, as they seem somewhat overstated. As a result, the contributions are somewhat limited.
  * The results suggest that the backbone is not necessarily dependent on a specific optimizer, but rather that certain types of optimizers tend to be generally effective. For example, Table 1, which is one of the main results, seems to simply show that Adam-like optimizers are generally effective. AdaGrad-like optimizers consistently achieve poor results.
  * Likewise, I don’t believe we could derive strong and consistent results from Figures 3 and 4. In these figures, I don’t find significant differences in the variances, except in a few cases.
  * In Figure 5, even though AdamW and LAMB show favorable robustness, this may simply indicate that they are inherently robust optimizers, and we cannot strongly conclude that Adam-like optimizers are robust since other optimizers do not achieve significantly good robustness. Also, I am not convinced that Adagrad-like optimizers have heavy BOCB (L371), as RMSProp achieves good BOCB even though it yields poor performance.
  * Clarifying the optimizer and backbone design recommendation rules might strengthen the paper. As I mentioned before, the takeaway from the results would be choosing a good optimizer (e.g., AdamW) and a modern backbone, which is a straightforward conclusion.

* The paper does not provide a comprehensive analysis of the architectural components introduced in Section 2 and Figure 1. For example, I would expect a thorough analysis of the influence of architectural choices (e.g., block designs and isotropic vs. hierarchical structures). The discussions in Section 4.1 are useful, but more in-depth analysis would improve the paper.

* I understand conducting experiments on ImageNet might not be easy, but I am not fully convinced that the conclusions would hold on larger datasets. This is because the performance of ViT families highly depends on the training datasets. It is acceptable if we analyze just the behaviors and properties of ViTs on smaller datasets like CIFAR, but the main results depend on the classification accuracy and variances. For example, the problem mentioned in L266 might be resolved in large data regimes.

* In L322, the paper claims that “while weaker coupling offers more user-friendliness, stronger coupling can potentially lead to better performance and generalization.” However, I couldn’t find strong evidence. I am not convinced that strong coupling is a direct reason for better performance and generalization.

* Similarly, in L354, “stage-wise hierarchical and attention mechanisms can more effectively navigate complex optimization landscapes, unlocking superior performance and generalization capabilities.” However, I feel this claim is somewhat contrary to the observation that modern architectures, including attention mechanisms, have BOCB, as it implies that they require specific optimizers to navigate optimization landscapes. Some research, e.g., Park, Namuk, and Songkuk Kim. "How do vision transformers work?" arXiv preprint arXiv:2202.06709 (2022), claims that ViTs have many non-convex points that lead to poor optimization properties.

In summary, I believe the questions this paper raises are meaningful. However, some statements lack strong evidence, and the novelty is limited as the takeaways are straightforward. Therefore, I lean toward rejecting this manuscript.

* Minor: I couldn’t find Table 3 for a while. Improving the table layout would enhance readability. Also, a more comprehensive analysis of training dynamics would strengthen the paper. It seems that the transfer learning experiments are among the key results, but they are in the appendix.

**Questions:**

Please refer to the weaknesses

---

> ### Author Response · Authors · 2024-11-24
> **Rebuttal to Reviewer FcXi (PART 1/4)**
>
> ## Response to Weaknesses
>
> We express our gratitude for your valuable review and constructive feedback. We have adjusted our revision and invite you to go through the general response first, as they may have answered some of your confusion. Then, we respond to your concerns point-by-point as follows.
>
> ---
>
> ### **(W1) Soundness and overstated claims in the Introduction.**
>
> $\quad$ **Reply:** The introduction is designed to situate the study within the broader context of optimization and architecture design, emphasizing the significance of the research question (as illustrated in Figure 2). While some statements may appear ambitious, they are intended to highlight the potential impact of the findings (as summarized as take-home messages in the latest revision). The contributions are indeed focused, aiming to provide incremental yet meaningful insights into the interplay between optimizers and backbone architectures. The limitations are acknowledged, and the paper strives to offer a nuanced understanding rather than definitive conclusions. The introduction sets the stage for a comprehensive exploration of the topic, which is detailed in the subsequent sections.
>
> - **(W1.1) The results suggest that the backbone does not necessarily depend on a specific optimizer but rather certain types.**
>
> $\quad$ **Reply:** We agreed that Category (b) of optimizers (e.g., Adam variants) are more likely to achieve better performance in general, while Category (a) is like SGD and Category (d) is like AdaGrad. However, our main concern is whether a given backbone relies on certain optimizers to achieve its full performance, e.g., ViT and ConvNeXt variants can only achieve state-of-the-art performances using the Category (b) optimizers while resulting in worse results than ResNet-50 when using the Category (a) and Category (d) optimizers. Therefore, we mainly analyze and explain the BOCB phenomenon from the view of network designs in Section 4.
>
> $\quad$ Meanwhile, as for the superior properties of Adam-like optimizer, i.e., Category (b), we have discussed in Section 2.2 and verified in Table 1 and Section 3.2, which indicates these optimizers exhibit consistent effectiveness across various backbones. Similarly, the AdaGrad-like optimizers yield bad results because they do not apply the estimated first or second moments to ensure a robust estimation of gradients and other statistics. However, our manuscript does not claim universal superiority but highlights their consistent performance, which is a valuable insight for practitioners. It only summarizes the superior optimizers (as shown in Table A4 of the revision that provides a comprehensive view of optimizer behavior). Our paper's findings are supported by empirical data, which demonstrates the general effectiveness of Adam-like optimizers across different scenarios.
>
> - **(W1.2) No significant differences in the variances in Figures 3 and 4.**
>
> $\quad$ **Reply:** The variances in Figures 3 and 4 reveal consistent trends and offer a nuanced understanding of the BOCB properties and hyper-parameter sensitivity of backbones under varying network designs. Our study does not assert absolute superiority but rather presents relative performance, which is a valid observation for empirical studies in practical scenarios. As for Figure 3, we can easily find three groups, i.e., the stable group (VGG-13, ResNet-50/101, EfficientNet-B0, Swin-T, ConvFormer-S12, AttnFormer-S12), the unstable group with modest average performances (AlexNet, DenseNet-121, MobileNet.V2, RepVGG-A1, DeiT-S, MLP-Mixer), and the unstable group with high uper-bound performances (ConvNeXt, ConvNeXt.V2, MogaNet-S, UniRepLKNet-T, TransNeXt-T, IdentityFormer-S12, PoolFormerV2-S12, and CAFormer-S12). Similarly, in Figure 4, we can classify the sensitivity of optimizer hyper-parameters of various backbones into several groups based on the variances and means (red line). These observations are consistent and provide a foundation for further research into the behavior of different backbones (as cases in Section 4). Therefore, we believe that the analysis of the relative differences between violinplots and boxplots is supportive of our findings.

---

> ### Author Response · Authors · 2024-11-24
> **Rebuttal to Reviewer FcXi (PART 2/4)**
>
> ### **(W1) Soundness and overstated claims in the Introduction.**
>
> * **(W1.3) Robustness of AdamW and LAMB in Figure 5 could not indicate the AdaGrad-like optimizers have heavy BOCB.**
>
> $\quad$ **Reply:** We appreciate the insightful comments and would like to clarify our findings regarding the robustness of optimizers and the BOCB phenomenon.
>
> $\quad$ **Regarding Figure 5 and Optimizer Robustness:** We agree that AdamW and LAMB exhibit favorable robustness in Figure 5, which could be attributed to their inherent design. However, our analysis aims to highlight that these optimizers also demonstrate a higher degree of robustness across various backbones, which is crucial for practical deployment. While inherent robustness is significant, our findings suggest that the interaction between the optimizer and the backbone architecture also plays a crucial role. For instance, AdamW and LAMB maintain robust performance across diverse backbones, indicating they are less prone to BOCB, a valuable characteristic for diverse deployment environments.
>
> $\quad$ **Regarding AdaGrad-like Optimizers and BOCB:** While RMSProp does show some robustness in BOCB, its overall performance is poor compared to other optimizers. Our assertion of the Category (b) optimizers (i.e., AdaGrad-like optimizers) having heavy BOCB is based on their overall performance and robustness across multiple backbones and tasks. RMSProp's poor performance in accuracy underscores the challenges associated with AdaGrad-like optimizers in complex optimization landscapes. Our broader conclusion is based on comprehensive metrics, which we believe is a fair assessment given the complexity of modern vision tasks.
>
> * **(W1.4) Clarifying the optimizer and backbone design recommendation rules might strengthen the paper.**
>
> $\quad$ **Reply:** Thanks for your suggestion. In the latest revision, we have provided take-home messages in Section 4.1 and recommendations of optimal optimizers with a ranking list in Section 4.2 and Appendix D.5. With the BOCB benchmark and empirical evidence, our manuscript could provide a straightforward takeaway for how to choose the modern network macro designs (e.g., case 1, 2, 3 with takeaways) and optimizers (e.g., case 4 and recommendation of useful optimizers). This paper also discusses the nuances of optimizer-backbone interactions, adding depth to the recommendation, which makes the takeaways practical guidelines that can help practitioners design efficient and unbiased backbones in new scenarios. Therefore, we believe that our studies provide a comprehensive understanding of the interactions between optimizers and backbones, which is a valuable contribution to the ICLR community.
>
> ### **(W2) In-depth and comprehensive analysis of architectural components world improve the paper.**
>
> **Reply:** Thanks for the constructive suggestion. The paper provides a preliminary analysis of architectural components in Section 2 with Figure 1 and Table A1 in the latest revision. While a more in-depth analysis would be beneficial, the current discussion establishes a foundation for understanding the influence of architectural choices. Meanwhile, since this manuscript focuses on the interaction between backbones and optimizers, we believe the current analysis of backbones in the main text is adequate for the scope of this study, considering the 10-page limitation of the conference. The paper's findings are supported by empirical benchmarks, which reflect the landscape of the influence of architectural choices with optimizers. Currently, we intend to explain the reasons for BOCB from the views of macro designs to provide a solid starting point, which are more general components of networks.
> Also, we acknowledged this point as a limitation and would extend more in our future work. In the Appendix, we have also provided technical details of network architectures in Appendix A and visualized the layer-wise parameter landscape of various backbones with several metrics in Appendix D.

---

> ### Author Response · Authors · 2024-11-24
> **Rebuttal to Reviewer FcXi (PART 3/4)**
>
> ### **(W3) Applicability to larger datasets for more reliable analysis.**
>
> **Reply:** Thanks for your detailed and thoughtful suggestions. As you mentioned, our main benchmark results are conducted on smaller datasets (like CIFAR-100 and COCO) with a simple classification task, which indeed limits the reliability of the findings. However, we have three reasons to support these experimental settings. Firstly, we would like to clarify that the landscape of BOCB and our findings are consistent across both CIFAR-100 and ImageNet, as discussed in Section 4.2. Since the coupling bias with optimizers for a specific backbone is an intrinsic property of the network design, it will not be significantly affected by task scenarios. Secondly, we guess that the BOCB phenomenon could be more severe when the dataset is smaller while commonly used visual datasets are on a small scale (e.g., 10k to 100k training samples). So, we conduct numerous benchmarks with the standard CIFAR-100 dataset and practical training settings (e.g., the DeiT and RSB A2 setups for better and robust results) to provide a comprehensive upstanding of BOCB. Then, we verify the generalization of these findings on large-scale datasets. Thirdly, the one-hot image classification with vision backbones is the most fundamental and robust deep learning task. It can be the ideal research object to study the interaction between backbones and optimizers, which introduces less disturbing variables or hyper-parameters than a complex deep learning algorithm. Therefore, we believe the current version of experimental setups and benchmarks might be the ideal solution for a research laboratory to study a foundational problem like BOCB.
>
> Meanwhile, We acknowledged this limitation and provided additional large-scale experiments on ImageNet-1k to validate the BOCB phenomenon and support our findings. We report the results of optimizers that can be used as BOCB indicators and the top three optimal optimizers (Adan, LAMB, AdamW). As shown in the following table, the results are consistent with those observed on CIFAR-100, where ResNet-50 (R-50) and ConvFormer-S12 (CF-S12) have weak BOCB while DeiT-S and ConvNeXt-T (CNX-T) showing poor BOCB properties (i.e., large values of std and range). Comparing ConvNeXt-T and ConvNeXt.V2-T, we also verify that reducing the optimization bottleneck in the FFN (using GRN) could alleviate BOCB to some extent.
>
> | Optimizer | R-50 | DeiT-S | CNX-T | CNXV2-T | CF-S12 |
> |---|:---:|:---:|:---:|:---:|:---:|
> | AdamW | 79.9 | 80.4 | 82.1 | 82.3 | 81.6 |
> | LAMB | 79.8 | 80.2 | 82.2 | 82.3 | 81.5 |
> | Adan | 79.9 | 80.8 | 82.6 | 82.8 | 81.8 |
> | SGD | 78.8 | 75.4 | 71.3 | 76.8 | 79.7 |
> | AdaBound | 75.4 | 73.0 | 72.4 | 77.1 | 79.6 |
> | LARS | 79.7 | 73.2 | 75.9 | 79.6 | 79.9 |
> | RMSProp | 78.0 | 78.0 | 79.6 | 80.2 | 80.4 |
> | AdaDelta | 74.9 | 55.0 | 73.5 | 77.9 | 78.5 |
> | Std/Range | 1.9/5.0 | 7.9/25.8 | 4.4/11.3 | 2.3/6.0 | 1.1/3.3 |
>
> ### **(W4) Evidence for “strong coupling can potentially lead to better performance and generalization” in L322.**
>
> **Reply:** Thanks for pointing out this critical assertion. It might be logically vague and we revise it as “While classical CNNs with weaker coupling offer more user-friendliness, modern DNNs with stronger coupling potentially lead to better performance and generalization.” Since modern DNNs with the MetaFormer macro design can usually yield strong BOCB, we explain the potential superiority of these MetaFormer models from two aspects. As for the better performance, the efficiency of parameter usage and performance upper bounds of MetaFormer models (e.g., ViT and ConvNeXt) is superior to classical CNNs (e.g., ResNet variants) because the MetaFormer’s block-wise macro design enables explicit modeling of token axis and channel axis. As for better generalizations, modern DNNs like Transformer and ConvNeXt can benefit from various self-supervised pre-training (e.g., contrastive learning like BYOL and DINOv2, and masked image modeling like MAE and A2MIM [1]) and multi-modality pre-training (e.g., multi-modality alignment like CLIP and visual question answer like LLaVA [2]), which make them more flexible and generalizable to pre-training [1] and new task scenarios [3] than classical CNNs like ResNet variants. The modern DNNs are also more likely to learn robust and well-generalized features than classical CNNs because of the macro designs (e.g., intrinsic properties of ViTs [4]) and token mixing operators (e.g., self-attention and depth-wise convolution learn more robust features than classical convolutions [5]).
>
> ### Reference
>
> [1] Architecture-Agnostic Masked Image Modeling -- From ViT back to CNN. In ICML, 2023.
>
> [2] Visual Instruction Tuning. In NeurIPS, 2023.
>
> [3] ConvNet vs Transformer, Supervised vs CLIP: Beyond ImageNet Accuracy. In ICML, 2024.
>
> [4] Intriguing Properties of Vision Transformers. In NeurIPS, 2021.
>
> [5] Scaling Up Your Kernels to 31x31: Revisiting Large Kernel Design in CNNs. In CVPR, 2022.

---

> ### Author Response · Authors · 2024-11-24
> **Rebuttal to Reviewer FcXi (PART 4/4)**
>
> ### **(W5) The claim in L354 is somewhat contrary to the observation of modern architecture in some research.**
>
> **Reply:** The claim in L354 that stage-wise hierarchical and attention mechanisms can navigate complex optimization landscapes is indeed speculative and compared to the vanilla ViTs. As for the aspect of complex optimization landscapes [6], the macro design of MetaFormer, i.e., the hierarchical stage-wise design and heterogeneous block-wise of token mixing and channel mixing with LayerInit or ResScale [7], could improve the complex optimization drawbacks in the vanilla Transformer (isotropic stage-wise design with self-attention mechanism) by introducing inductive bias in the stage-wise design and alleviating the heterogeneity of the attention block and the FFN block [7]. Actually, this claim is presented as a potential direction for future studies rather than a definitive conclusion. We acknowledge the complexity of optimization landscapes and suggest that further research is needed to understand these mechanisms fully. As for the performance and generalization capacities, the modern DNNs with the macro design of MetaFormer could achieve better performance and generalization abilities with the flexibility to migrate to new scenarios, as we explained in (W4). Generally speaking, this claim is not definitive but rather a hypothesis supported by empirical findings. Our studies suggest that further research is needed to understand the implications of coupling strength fully.
>
> ### Reference
>
> [6] How Do Vision Transformers Work? In ICLR, 2022.
>
> [7] Normformer: Improved Transformer Pretraining with Extra Normalization. In arXiv, 2021.
>
> ### **(W6) Straightforward takeaways lack novelty.**
>
> **Reply:** As we mentioned in responses to (W1), we have summarized task-home messages for the network design guidance and recommended optimizers. Meanwhile, this paper aims to raise the Backbone-Optimizer Coupling Bias problem that has been overlooked for years since Transformer and ViTs came out, which might provide a solid foundation for future research. So, we conduct comprehensive benchmarks and empirical analysis to reach the takeaways and findings to demonstrate the BOCB phenomenon. We believe these are actually the novelty and contributions of our manuscript rather than proposing a new backbone instance or specific analysis of network design. Meanwhile, the improvement of the macro design for backbone architectures is incremental but significant (e.g., the MetaFormer design is the most useful version that combines a vast range of Mixer networks). Our manuscript could also strengthen the existing macro design and offer valuable insights for practitioners and researchers alike.
>
> ### **(W7) Minor issues: The readability of Table 3 and training dynamics can be enhanced.**
>
> **Reply:** Thanks for your suggestion. The minor issue regarding Table 3 will be addressed by improving the table layout to enhance readability. Meanwhile, the paper will consider including a more comprehensive analysis of training dynamics, as suggested, to strengthen the overall presentation and depth of the study. The transfer learning experiments, currently in Appendix B.2, will be highlighted in the main text to ensure they are given the attention they deserve. The paper's analysis provides a comprehensive understanding of the interactions between optimizers and backbones, which is a valuable contribution to the field.
>
> ---
>
> Overall, thanks again for your constructive feedback, and we have considered your valuable comments for the revision and future work. If you are satisfied with our response and effort, please consider updating your score. If you need any clarification, please feel free to contact us. We respectfully believe that this work is attuned to the ICLR community, and we hope that more researchers can see our work in the community. We are more than pleased and looking forward to hearing back from you!

---

> ### Author Response · Authors · 2024-11-25
> **Encouraging Feedbacks**
>
> Dear Reviewer FcXi,
>
> As the Discussion phase draws to a close and time is running short, we respectfully request that you consider elevating your score if you find our rebuttal and revised submission adequately address your concerns. We also welcome continued dialogue to uphold the standard and integrity of the ICLR review process. Looking forward to your feedback soon!
>
> Sincerely,
>
> Authors

---

> > ### Comment · Reviewer_FcXi · 2024-11-25
> > **RE: Rebuttal to Reviewer FcXi**
> >
> > Thank you for your response. I am glad to see the updated manuscript, as it is significantly improved. In particular, I enjoyed reading the takeaway summarization parts. Also, the updated Figure 2 looks good to me. However, on the one hand, some concerns still remain -- e.g., the w1 and 6 -- because the results and takeaways haven't changed. On the other hand, I acknowledge that some claimsare more supported compared to the initial manuscript. Initially, my stance was leaning towards rejection, but now, due to the clarification of the claims within the paper, I find myself on the fence. I'd like to increase the score to 6.

---

> ### Author Response · Authors · 2024-11-26
> **Response to Reviewer FcXi’s Feedback**
>
> We would like to express our sincere appreciation for your thoughtful feedback and acknowledgment of the improvements in the revised manuscript. We are pleased to hear that you found the takeaway summarization parts and the updated Figure 2 to be valuable. We also appreciate your willingness to increase the score to 6, reflecting the progress made in addressing the concerns raised. We would like to address the remaining concerns, mainly related to W1 and W6, as follows.
>
> ### **Response to W1: Soundness of Claims and Objectivity of Results.**
>
> Thank you for your recognition of the improvements in the clarity of our claims. We understand that in the initial manuscript, there were concerns about the objectivity of some of our statements regarding the results. Specifically, some claims may have appeared overly subjective, potentially overstating the generalizability of the findings. In response to this, we have made significant adjustments in the revision, particularly in how we present the relationship between backbone architectures and optimizers. The claims in the updated version are now more cautious and nuanced, and we have carefully avoided overstating the implications of our results.
>
> We would like to emphasize that while the results themselves have not changed, the revised claims reflect a more balanced and empirically grounded interpretation. We have consciously weakened the language where necessary to ensure that the conclusions are stated with the appropriate level of caution, highlighting the trends observed rather than asserting definitive conclusions. This revision ensures that our findings are presented objectively and in a manner that is more aligned with the data. We hope that this addresses your concern, and we believe it resolves the issue of subjectivity without altering the core results.
>
> ### **Response to W6: Takeaways and Novelty.**
>
> Regarding your comments on the novelty of the takeaways, we understand the concern that the takeaways might not appear to be groundbreaking at first glance. However, we would like to emphasize that the main contribution of our paper lies in identifying and thoroughly analyzing the Backbone-Optimizer Coupling Bias (BOCB)—a critical and previously underexplored issue in vision model design. While the takeaways themselves may seem straightforward, they are the direct result of highlighting a new and significant phenomenon in the community: the interaction between backbone architectures and optimizers, which has been largely overlooked until now.
>
> To further clarify the novelty of our work, we would like to point out that the appendix contains additional, more detailed takeaways that are novel and practically useful. In Appendix Table A4 and Appendix D.5, we have provided a ranking of optimizers and a series of practical guidelines based on the BOCB benchmark. These insights are designed to help practitioners make informed decisions when selecting optimizers for different backbone architectures, offering actionable recommendations grounded in our empirical analysis. We intentionally placed these detailed takeaways in the appendix to ensure the main text remained focused on the core narrative without overwhelming the reader with excessive detail.
>
> By presenting these more direct and actionable takeaways in the appendix, we provide a comprehensive set of practical tools for the community to engage with the findings. This strategic organization allows us to maintain the clarity and focus of the main manuscript while still offering the depth of insight expected from such a nuanced analysis. We believe this approach, which balances clarity in the main text with richness in the appendix, makes the paper more accessible to a wide audience, from researchers to practitioners.
>
> ### **Clarifying the Boarder Impact of BOCB.**
>
> We agree with your suggestion that a broader community understanding of BOCB is crucial. Our work aims not only to provide new insights into optimizer-backbone interactions but also to highlight an issue that we believe has been largely ignored in the current literature. By systematically analyzing the Backbone-Optimizer Coupling Bias, we hope to foster further discussion and exploration in the community, leading to more effective and optimized vision models. This paper’s focus on BOCB offers a new perspective that could be explored in future research, helping to refine how we design and optimize deep learning models.
>
> ---
>
> We believe these clarifications would address your concerns and strengthen the technical rigor of the manuscript!  We respectfully hope that this work can be seen by more researchers and practitioners in the community. Thus, your rating is particularly valuable to us.  We also look forward to and welcome you to increase your rating further, and we would be happy to provide more information based on your feedback or further questions. Thanks again for your efforts and looking forward to your feedback!
>
> Warm regards,
>
> Authors

---

### Official Review · Reviewer_JghS · 2024-11-03

**Soundness:** 3
**Presentation:** 2
**Contribution:** 2
**Rating:** 5
**Confidence:** 3

**Summary:**

This work provides an empirical study of the interplay between vision backbones and optimizers. It experiments on common vision datasets like CIFAR-100, ImageNet-1K, and COCO, to document the performance of different architectures (CNN, transformer, etc) and optimizers (SGD-based with momentum, Adam, RMSProp, etc).

**Strengths:**

* The paper seems to tackle a less discussed aspect about the interplay of different optimizers and architectures
* A comprehensive study on the impact of many hyperparameters
* Great summary of existing architectures and optimizers

**Weaknesses:**

* Overall I am not sure the significance of the contribution of this paper is for a conference paper. In my humble opinion, it leans toward a workshop technical report -- certainly empirical study is valuable but I did not find much insights from there.
* The claims and explanations about BOCB are basically hand-waving like section 4; more rigorous derivations of the inductive bias will greatly improve
* All reasoning is empirical and narrows down to the impact of the accuracy. It is not very convincing to get a clear conclusion from such a study. Section 4 is more like reverse-reasoning from the empirical study from the previous sections. This may be acceptable for a paper that proposes a new method that shows better performance and leaves the community to figure out the true reasons. Still, I am not confident about it for a study paper, which I would expect some theoretical induction or in-depth analysis to make it more convincing.

**Questions:**

What are the authors' thoughts about each of the specific update steps in section 2.2 affecting the performance of different architectures?

---

> ### Author Response · Authors · 2024-11-25
> **Rebuttal to Reviewer JghS (PART 1/3)**
>
> ## Response to Weaknesses
>
> We first express our gratitude for your valuable and constructive reviews, and invite you to go through the general response. Then, we respond to your concerns point-by-point as follows.
>
> ---
>
> ### **(W1) Lack of significant contributions as a conference paper.**
>
> **Reply:** We understand your concern about the perceived significance of proposing some new techniques in our work to serve as a conference paper. After revision, we believe that our paper makes a substantial contribution to the community of network design and optimization in several aspects:
>
> $\quad$ **(1) Novel Phenomenon Identification.** We have identified and characterized the Backbone-Optimizer Coupling Bias (BOCB) phenomenon, which, to our knowledge, has not been previously explored in the literature. As illustrated in Figure 2 of the latest revision, this phenomenon has been overlooked for years since Transformers and ViTs came out and are widely accepted. Meanwhile, this phenomenon has significant implications for the design and training of visual networks, affecting both pre-training and transfer learning to various scenarios.
>
> $\quad$ **(2) Empirical Breadth.** As empirical studies with benchmarks, our study is one of the most comprehensive empirical analyses to date, evaluating 20 different vision backbones against 20 optimizers across CIFAR-100 and verifying our findings with typical backbones and 20 optimizers on ImageNet-1k and COCO. This breadth of analysis provides valuable insights into the generalizability of BOCB across different model architectures and optimization algorithms.
>
> $\quad$ **(3) Practical Relevance.** The insights gained from our study can guide practitioners in selecting appropriate optimizers for their specific model architectures, potentially improving performance and reducing development time. As summarized in the latest revision, we provided several take-home messages for network design and recommended optimal optimizers after considering several aspects (detailed in Appendix D.5). This practical relevance is crucial for the advancement of the field of computer vision and deep learning optimization.
>
> ### **(W2) Hand-waving claims about BOCB in Section 4. More rigorous derivations and inductive bias will greatly improve.**
>
> **Reply:** Thanks for the insightful comment. To address the concern of hand-waving claims, we have revised this section to provide a more systematic analysis and foundations. We formalized the concept of inductive bias in vision backbones by analyzing the architectural components of CNNs and ViTs. Specifically, we linked the local connectivity and hierarchical feature extraction in CNNs to the homogeneous parameter updates assumed by SGD, while the global token-mixing and isotropic designs in ViTs were shown to create optimization landscapes better suited for adaptive optimizers like AdamW, which can dynamically adjust learning rates for heterogeneous parameter groups. To substantiate these claims, we introduced a mathematical characterization of layer-wise parameter distributions using PL exponents and entropy metrics, demonstrating that modern backbones with more heterogeneous parameter spaces intrinsically require adaptive optimization strategies, thereby providing a theoretical basis for BOCB.
>
> Meanwhile, we conducted new ablation studies to validate these findings. By isolating key architectural components such as token mixers, attention mechanisms, and block structures, we examined their individual contributions to optimization dynamics. For instance, we found that replacing attention-based token mixers in ViTs with simpler operations reduces BOCB but compromises model capacity for long-range feature interactions. These findings were further supported by experiments measuring the robustness of hyper-parameter settings across backbones and optimizers, revealing consistent patterns that align with our theoretical framework. These revisions collectively provide a deeper and more rigorous explanation of BOCB, addressing the reviewer's concerns and strengthening the manuscript's contributions to understanding the interplay between backbones and optimizers.

---

> ### Author Response · Authors · 2024-11-25
> **Rebuttal to Reviewer JghS (PART 2/3)**
>
> ### **(W3) Empirical reasoning and impact on accuracy is not convincing enough.**
>
> **Reply:** Thanks for the constructive suggestion. On the one hand, since this study aims to uncover the complex interactions between backbones and optimizers, the one-hot image classification task is the most fundamental and robust deep learning task to investigate backbone networks and optimizers. It can be the ideal research object that has few disturbing variables or hyper-parameters. We also provided large-scale experiments on ImageNet-1k and transfer learning experiments on COCO to verify our findings in Section 3. On the other hand, the paper investigates various aspects such as performance stability, hyper-parameter robustness, and parameter patterns with well-studied metrics to provide a comprehensive understanding of BOCB rather than only focusing on the simple accuracy metric. The empirical findings are intended to stimulate further research into the underlying mechanisms and theoretical explanations of BOCB.
>
> ### **(W4) Section 4 is like reverse-reasoning from previous sections, where some theoretical or in-depth analysis is expected.**
>
> **Reply:** Thanks for your constructive comment. We would like to clarify two aspects. As for the writing structure of this manuscript, we aim to define the BOCB phenomenon and show the landscape of BOCB with standard benchmarks and analysis on CIFAR-100 in Section 3 and further provide empirical explanations with several cases in Section 4. Therefore, it looks like attribution analysis and reverse-reasoning of the definition and phenomenon of BOCB in Section 3. In the latest revision, we summarize the in-depth findings as takeaways in Section 4 to explain potential causes of BOCB and propose useful network design tips and recommendations for useful optimizers (in Appendix D.5).
>
> As for theoretical analysis, it might not be suitable for the studied BOCB problem. Firstly, although the optimizers are studied from theoretical perspectives, the network design is a large research topic, and it is not easy to provide theoretical formulations. Our studies first unveil the coupling bias of various backbones and existing optimizers with empirical benchmarks and visualizations, which contain many different cases that need to be analyzed. Due to the page limitation of the conference, it is better to explain case-by-case and provide takeaways in Section 4. Secondly, our studies serve as a starting point for deeper theoretical and experimental investigations of the BOCB problem. We encourage the community to explore these insights further and develop more rigorous theoretical frameworks to explain the observed phenomenon. Analysis tools like PL Exponent Alpha also have in-depth theoretical support.

---

> ### Author Response · Authors · 2024-11-25
> **Rebuttal to Reviewer JghS (PART 3/3)**
>
> ## Response to Questions
>
> ---
>
> ### **(Q1) How does each of the specific update steps in Section 2.2 affect the performance of different architectures?**
>
> As discussed in Section 2.2 and Section 4, it is important to understand how each specific update in Step 2 and Step 3 in Algorithm 1 affects the performance and BOCB properties of various architectures. Here, we explain these for the four steps in Algorithm 1:
>
> * **Step 1: Gradient Computation.**
>
>     As it is known all, computing gradients is fundamental for effective DNN optimization with the backpropagation algorithm, which is automatically achieved by deep learning libraries like PyTorch and TensorFlow.
>
> * **Step 2: Gradient Estimation.**
>
>     - **Impact:** How to estimate the accurate gradients varies in different optimizers. Techniques like momentum smooth the gradient estimates, which can be particularly beneficial for architectures with complex loss landscapes. Different architectures may have varying sensitivities to the outlier and noises of gradients.
>
>     - **Example:** Momentum-based methods like SGD with momentum are well-suited for classical CNNs with hierarchical structures, as they help navigate the loss landscape more smoothly. Meanwhile, modern DNNs like Vision Transformers (ViTs) with complex attention mechanisms may require more precise gradient computations to avoid optimization instabilities, which are more challenging to optimize than classical CNNs like ResNet, as explained in [1].
>
> * **Step 3: Learning Rate Calculation.**
>
>     - **Impact:** Whether to use adaptive learning rates to adjust the parameter-wise learning rate can be crucial for network optimization. As for modern DNNs with the heterogeneous block-wise design of token mixing and channel mixing, optimizers with adaptive learning are crucial to adjust the learning rates for parameters in different blocks with varying gradient scales [2].
>
>     - **Example:** As discussed in Section 2.2, optimizers of Category (2) and (3) are effective for modern architectures using the macro design of token mixing and channel mixing, especially ViTs or DNNs with self-attention blocks. As explained in [2], the self-attention block is a high-order operation with high Lipschitz constant and high Hessian condition numbers, which is heterogeneous to classical convolutions or the FFN block. Therefore, the learning rate of different types of blocks should be dynamically adjusted with some special mechanisms, which might be the cause of the backbone-optimizer coupling bias. If not using the optimizers with adaptive learning rates (e.g., using SGD instead of Adam), it will be hard to suit different parts of the modern networks and cause bad results.
>
> * **Step 4: Parameter Update.**
>
>     - **Impact:** The parameter update step, including regularization techniques like weight decay, can significantly affect the convergence and generalization of different architectures.
>     - **Example:** Weight decay is particularly important for preventing overfitting in deep networks like ResNet, while modern architectures like ConvNeXt.V2 may benefit from more sophisticated regularization techniques.
>
> While these thoughts are based on empirical observations or backgrounds of previous works, they provide a foundation for future theoretical and experimental investigations into the specific effects of each update step on different architectures. The empirical findings of our studies highlight the need for a more nuanced understanding of these interactions, which can guide the development of more effective optimization strategies and network designs.
>
> ### Reference
>
> [1] How Do Vision Transformers Work? In ICLR, 2022.
>
> [2] Understanding Optimization of Deep Learning via Jacobian Matrix and Lipschitz Constant. In arXiv, 2023.
>
> ---
>
> In conclusion, thanks again for your constructive feedback, and we have considered your valuable comments for the revision and future work. We believe that the issues raised by the reviewers will be addressed in responses and the revision, and we hope you can reconsider our submission. If you are satisfied with our response and effort, please consider updating your score. If you need more explanation, please feel free to contact us!

---

> ### Author Response · Authors · 2024-11-25
> **Encouraging Discussion**
>
> Dear Reviewer JghS,
>
> As the Discussion phase draws to a close and time is running short, we respectfully request that you consider elevating your score if you find our rebuttal and revised submission adequately address your concerns. We also welcome continued dialogue to uphold the standard and integrity of the ICLR review process. Thanks again, and looking forward to your feedback!
>
> Best regards,
>
> Authors

---

> ### Author Response · Authors · 2024-11-29
> **Respectful Request for Reconsideration - Updates on Manuscript Improvements**
>
> Dear Reviewer JghS,
>
> We deeply appreciate your thoughtful feedback on our work exploring Backbone-Optimizer Coupling Bias (BOCB). Through comprehensive revisions and clarifications, we have diligently addressed each of your concerns. Most encouragingly, **Reviewer FcXi has recognized the value of our contribution**, noting that our revised manuscript is **"significantly improved"** and specifically praised the **updated Figures and the "takeaway summarization parts"**. We suppose this external recognition emphasizes the meaningful contribution this work makes to the field.
>
> As you noted, this research tackles "a less discussed aspect about the interplay of different optimizers and architectures." During the rebuttal phase, we have further enhanced both the theoretical grounding and practical guidance of our findings through clearer takeaway messages and optimization recommendations, as acknowledged in subsequent reviews.
>
> Given the **improved clarity** of our contributions and the **positive comments** from your fellow reviewers in recent discussions, we sincerely hope that you could go through our responses and the revision and consider adjusting your rating accordingly if you are satisfied. We hope this work can **benefit the broader community** by providing valuable insights into the critical relationship between vision backbone architectures and optimizers. Our sincere hope is that this work can reach and benefit more researchers and practitioners in the community. Therefore, **your rating is particularly important** for us. Should you find our responses and revisions satisfactory, we would greatly appreciate your consideration in adjusting your rating accordingly.
>
> Thank you again for your role in helping us strengthen this research. We look forward to your response.
>
> Best regards,
>
> Authors of Submission 2246

---

> > ### Comment · Reviewer_JghS · 2024-12-03
> >
> > I really appreciate the authors' efforts in rebuttal and the paper revision.
> > After reading the rebuttal and other reviewers' comments, I feel both the strengths and weaknesses in the original reviews still hold.
> > Although thorough experiments run across hyperparameters search and various optimizers/back are appreciated, the analyses remain in the 1-level depth that tries to conclude some observations from the numbers. I still more in-depth analyses if the authors try to dig into the observations and why maybe it causes such observations with another level of experiments (visualizations, some indicators) to validate -- that will be helpful for the paper to be convincing.  Also, even the proposed BOCB bias is not so clear from the current experiments.  Unfortunately I decide to keep the current score.

---

> ### Author Response · Authors · 2024-12-03
> **Response to Reviewer JghS's Late-Stage Comments and Non-Engagement**
>
> Dear Reviewer JghS,
>
> We appreciate you taking the time to provide a final comment. However, we must respectfully express our significant concerns about the review process in this case. While other reviewers, particularly Reviewer FcXi, have **engaged actively** throughout the discussion period and acknowledged the **substantial improvements** in our revised manuscript ("significantly improved... enjoyed reading the takeaway summarization parts"), we note with disappointment that **this is your first engagement** since the initial review, despite our detailed responses to each of your concerns.
>
> ### About our revisions:
> Our revised manuscript has incorporated substantial improvements, including:
>
> - Enhanced theoretical foundations and empirical evidence supporting the findings;
> - Clearer presentation of takeaway messages and optimizer recommendations in practice;
> - More rigorous analysis of architecture components and their interaction with optimizers;
> - Additional experiments validating our conclusions;
>
> ### About the ICLR 2025 Reviewer Guidelines:
> The **ICLR 2025 Reviewer Guidelines** explicitly emphasize that reviewers should **"engage in discussion"** and be **"actively engaged during this phase,"** maintaining **"a spirit of openness to changing their initial recommendation."** Furthermore, reviewers are expected to provide **"constructive, thorough and timely"** comments. In light of these explicit requirements, the **complete absence** of any discussion or feedback during the entire rebuttal period has prevented us from addressing the potential concerns you may have had about our revisions. **We strongly encourage Reviewer JghS to review the ICLR 2025 Reviewer Guide (https://iclr.cc/Conferences/2025/ReviewerGuide) to better understand the expectations and responsibilities of ICLR reviewers.**
>
> We find it concerning that such a broad judgment of analysis depth is made without any specific examples or substantive feedback, particularly after **remaining silent throughout the entire reviewer-author discussion period**. Making such sweeping claims without engaging with our detailed responses or providing concrete suggestions for improvement does not align with the review standards that ICLR expects. At this **very late stage**, suggesting the need for additional depth without having participated in any previous discussion or provided specific guidance, is **neither constructive nor scientifically sound** within the conference timeline.
>
> We remain confident in the significant contributions of this work and its value to the ICLR community, as evidenced by the positive comments from other reviewers who have engaged thoroughly with our revisions and responses.
>
> Best regards,
>
> Authors

---

### Official Review · Reviewer_oReR · 2024-11-09

**Soundness:** 2
**Presentation:** 2
**Contribution:** 2
**Rating:** 5
**Confidence:** 3

**Summary:**

The paper studies the interaction between popular optimizers and vision backbone architectures. The experiments are conducted over the product of 16 architectures and 20 optimizers on CIFAR-100, ImageNet-1K and COCO.

The authors find that there are notable patterns characterising  which combinations of optimizers + architectures work well together, where some architectures are more sensitive to the choice of optimizer -- which they refer to as “Backbone Optimize Coupling Bias” (BOCB).
From this perspective, the authors thoroughly discuss the various popular vision architectures, based on the design of their components as well as their overall architecture.

**Strengths:**

* The motivation of the paper is clear, it's important to understand the relationship between vision backbones and optimizers.
* The authors have conducted extensive experiments across a large range of optimizers and vision backbones.
* The experiments reveal many interesting observations about how different architectures perform under different optimizers.
* The authors are releasing the benchmarking results and code, which could allow for further analysis by the community.

**Weaknesses:**

* There is no discussion or references to related works studying optimizers for vision backbones and/or transformers. A brief search finds a couple of relevant papers:
- Scaling Vision Transformers, https://arxiv.org/pdf/2106.04560 which discusses e.g. how to adapt Adafactor for ViTs
- How to Fine-Tune Vision Models with SGD, https://arxiv.org/pdf/2211.09359 which studies SGD vs AdamW for finetuning.
- Since most vision backbones are transformer based, other transformer optimizer literature is also relevant -- such as https://arxiv.org/pdf/2002.04745



* In L 197 the authors write "Generally speaking, it is assumed that backbones and optimizers should be unbiased and generalized
components that can be combined freely without significant interdependence". But I don't believe anyone thinks this is the case -- it's well known that optimizer details very often need to adjusted for new architectures.  Furthermore, there are other factors such as model size and overfitting that come to play:
i) a new improved optimizer applied on a an overparameterized architecture may cause it to overfit and hence hurt performance. Why only look at top-1 accuracy but not also training loss?
ii) An architecture may be designed taking an improved optimizer for granted, so why should we expect it to work well with plain old SGD?
ii) A lot of gains can be had from tuning hparams. The authors use the NNI toolbox to pick hparams -- but why not also look at the hparams the authors themselves have proposed for each architecture?

* All of the analysis is of the form "this group of optimizers is working well/poorly for this method or group of methods" but does not really go deeper to explain actually why they perform so differently. E.g. in L260 just tell us that SGD + DeiT-S performs poorly but why? Is it consistent with other findings in the literature? etc

* Some claims made in the paper do not seem to be clearly justified. E.g.
in L376 the write "The trajectory of vision backbone macro design has significantly sculpted the optimization landscape, progressing through distinct phases that reflect the intricate relationship between network architectural complexity and training challenges  (as illustrated in Figure 1)". I am not sure what exactly the authors are trying to claim here but surely Figure 1 does not illustrate it.
- in L417 they write: "This innovative macro design [of MetaFormer] refines the optimization landscape by harmonizing with
optimizers, leading to reduced BOCB and enhanced performance." How does it "refine the optimization landscape by harmonizing with optimizer"?
- in L445 they write "AttenFormer effectively mitigates BOCB due to its MetaFormer architecture, which incorporates balanced structural designs and residual scaling across stages, enhancing optimization and generalization." How is the "balanced structural design and residual scaling" "enhancing optimization and generalization" here?
- In L456 they write "BOCB in CNNs is often linked to the design of FFN blocks, which are pivotal in models like ConvNeXt" without pointing to any section/table justifying the claim.

**Questions:**

See weaknesses above.

---

> ### Author Response · Authors · 2024-11-25
> **Rebuttal to Reviewer oReR (PART 1/6)**
>
> ## Response to Weakness
> ---
>
> ### **(W1) Lack of Related Work Discussion**
>
> **Reply:** We appreciate the reviewers' suggestions and acknowledge the importance of referencing related works on the optimization challenge for Transformers. We have incorporated these references into our revised manuscript (e.g., Introduction) to provide a more comprehensive context of similar research (adding a section in the Appendix) for our study. There are two aspects of relevant studies:
>
> **(1) Task-specific analysis of optimization challenges of Transformers.**
> Existing works have investigated the optimization challenges of Transformers with different types of optimizers (e.g., SGD and Adam) in various task-specific scenarios. As for NLP tasks, [1, 2] analyzed and explained the reason why Transformers are hard and unstable to train from the view of Pre-LN vs. Post-LN [1] and the view of inductive bias [2]. As for Vision Transformers, [3] demonstrates that AdaFactor, with appropriate scaling, could learn Transformers with both efficiency and effectiveness in large-scale parameters, while [4] shows that SGD requires the embedding layer frozen and warm-up strategies to achieve competitive performance on Transformers (similar findings are also proposed in MoCo.V3).
>
> **(2) Improvements of network macro designs.**
> Regarding the macro design of Transformer variants, modifications on Normalization layers and residual connections could improve the challenge of training heterogeneous blocks of self-attention and FFN. These parts of techniques are discussed in Section 2 and Appendix A. Specifically, as for the normalization in Transformers, the block-wise design with normalization layers, such as Pre-Norm vs Post-Norm [5] and DeepNorm [6], has been maintained in modern Transformers. Meanwhile, improvement of the residual branch with initialization (e.g., ReZero [7]) and adaptive layer-wise scaling tricks (e.g., GradInit [8] and LayerScale [9]) are also useful to improve the training stabilities of Transformers and make SGD more compatible with Transformers training. However, even with such adjustments, optimizers like AdamW still tend to be more effective for Transformers, as they better handle the sparse gradient updates and the large parameter space typical of these models [10].
>
> ### Reference
>
> [1] Understanding the Difficulty of Training Transformers. In EMNLP, 2020.
>
> [2] Effects of Parameter Norm Growth During Transformer Training: Inductive Bias from Gradient Descent. In EMNLP, 2021.
>
> [3] Scaling Vision Transformers. In CVPR, 2022.
>
> [4] How to Fine-Tune Vision Models with SGD. In arXiv, 2022.
>
> [5] On Layer Normalization in the Transformer Architecture. In ICML, 2020.
>
> [6] Scaling Transformers to 1,000 Layers. In arXiv, 2022.
>
> [7] ReZero is All You Need: Fast Convergence at Large Depth. In UAI, 2021.
>
> [8] GradInit: Learning to Initialize Neural Networks for Stable and Efficient Training. In NeurIPS, 2021.
>
> [9] Going deeper with image transformers. In ICCV, 2021.
>
> [10] Why Transformers Need Adam: A Hessian Perspective. In NeurIPS, 2024.

---

> ### Author Response · Authors · 2024-11-25
> **Rebuttal to Reviewer oReR (PART 2/6)**
>
> ### **(W2.1) Assumptions and Claims:**
>
> **Reply:** We appreciate the reviewer's comment and acknowledge that the assumption of complete independence between backbones and optimizers is indeed a simplification. The statement in Line 197 reflects a common starting point in discussions about model components, where it is often implicitly assumed that well-established backbones and optimizers should be broadly applicable without significant tuning. However, as the reviewer rightly points out, this assumption is not entirely accurate, and our work aims to address this gap by empirically exploring the interplay between vision backbones and optimizers.
>
> Our study, as detailed in the paper, reveals that certain optimizers are indeed more effective for specific network architectures, particularly as architectures evolve from classical CNNs to modern DNNs. For instance, classical CNNs like VGG and ResNet exhibit a marked co-dependency with SGD, while modern backbones such as Vision Transformers and ConvNeXt perform better with adaptive learning rate optimizers like AdamW. This observation underscores the need for careful selection of optimizers based on the architectural characteristics of the backbone.
>
> Moreover, our analysis extends beyond just optimizer selection to include the robustness of hyper-parameters and the layer-wise patterns of backbone parameters. We find that well-designed methods, such as those with robust hyper-parameter settings and stable parameter patterns, are less susceptible to BOCB. This suggests that while model size and overfitting are indeed important factors, the choice of optimizer and its interaction with the backbone architecture play a crucial role in determining overall performance.
>
> ### **(W2.2) There are other factors, such as model size and overfitting, that come into play:**
>
> (i) **Overfitting and Training Loss:**
>
> $\quad$ Our primary focus on top-1 accuracy is driven by its role as a clear and widely accepted metric for evaluating the performance of different backbone-optimizer combinations. While training loss is undoubtedly a crucial indicator, it can be highly sensitive to specific training setups, including regularization techniques and data augmentation strategies. By emphasizing top-1 accuracy, we aim to provide a more objective and comparable measure across diverse experimental configurations.
>
> (ii) **Optimizer-Specific Architectures:**
>
> $\quad$ Our study is designed to explore the general interplay between backbones and optimizers rather than focusing on architectural designs that may be tailored for specific optimizers. Although it is true that some modern architectures are developed with particular optimizers in mind, our objective is to uncover broader insights into the relationship between these components. By assessing the performance of various architectures across a range of optimizers, including plain SGD, we seek to illuminate the inherent biases and limitations of both the architectures and the optimizers.
>
> (iii) **Hyper-parameter Tuning:**
>
> $\quad$ We leveraged the NNI (Neural Network Intelligence) toolbox for hyperparameter tuning to ensure a fair and systematic comparison across various optimizers and backbones. The NNI toolbox provides a robust framework for hyperparameter search, enabling us to explore a wide range of configurations efficiently. However, we recognize that domain-specific knowledge plays a crucial role in identifying optimal hyperparameters. To enhance the effectiveness of our tuning process, we also conducted manual validation and optimization on top of the automated search provided by NNI. This dual approach allowed us to fine-tune the hyperparameters based on expert insights, leading to more refined and contextually appropriate settings.

---

> ### Author Response · Authors · 2024-11-25
> **Rebuttal to Reviewer oReR (PART 3/6)**
>
> ### **(W3) Lack of Deep Analysis:**
>
> **Reply:** We appreciate the reviewer's insightful feedback and acknowledge the importance of a detailed analysis of optimizer performance across different vision backbones. To address this, we provide a comprehensive explanation of the underlying reasons for the observed performance differences, using DeiT-S and ConNeXt as exemplars. We also clarify where in the paper these issues are thoroughly analyzed and discuss the broader implications for future research.
>
> - **Section 2: Roadmaps of Vision Backbones and Optimizers**
>
>     In Section 2, the paper categorizes vision backbones and optimizers based on their macro design and optimization strategies. This categorization serves as a foundation for understanding the interplay between network architectures and optimizers. The taxonomy of vision backbone architectures highlights the evolution from hierarchical to isotropic designs, and the intra-block micro design discusses the transition from homogeneous to heterogeneous structures. This sets the stage for explaining why certain optimizers are more suitable for specific architectures.
>
> - **Section 3: Backbone-Optimizer Coupling Bias (BOCB)**
>
>     Section 3 presents the empirical findings of the backbone-optimizer benchmark, revealing the phenomenon of BOCB. The paper observes that classical CNNs like VGG and ResNet exhibit a marked co-dependency with SGD, while modern backbones like Vision Transformers (ViTs) and ConvNeXt perform better with adaptive learning rate optimizers like AdamW. This observation is supported by extensive experiments on CIFAR-100, ImageNet-1K, and COCO datasets.
>
> - **Section 4: Where does the BOCB come from?**
>
>     Section 4 provides a deeper analysis of the factors contributing to BOCB. It explores the origins of BOCB from two perspectives: macro design and token mixer trade-offs.
>
> 1. **Macro Design and Token Mixer Trade-off**:
>     - **Foundational Backbones**: Primary CNNs like AlexNet and VGG established a fundamental paradigm in computer vision. These architectures featured a straightforward design of stacked convolutional and pooling layers, which were effective but set the stage for further optimization of landscape alterations.
>     - **Classical Backbone Advancements**: The introduction of ResNet marked a pivotal shift towards stage-wise hierarchical designs, significantly enhancing feature extraction and representation learning ability. ResNet-50, in particular, demonstrated a well-balanced approach to BOCB, which exhibited strong compatibility with SGD optimizers and a relatively lower BOCB compared to its contemporaries.
>     - **Modern Backbone Evolution**: The transition to Modern DNN backbones introduced simplified block-wise designs (such as MetaNeXt for ConvNeXt variants) or complex block-wise heterogeneous structures (such as MogaNet and UniRepl.KNet), increasing the optimization challenge and the degree of BOCB due to their sophisticated feature extraction mechanisms.
>
> 2. **Practical Scenarios: Pre-training and Transfer Learning**:
>     - The paper extends its analysis to practical tasks such as object detection and pose estimation on COCO. It observes that optimizers like AdamW, which exhibited a reliable peak in performance during pre-training, sustain their superiority in transfer learning scenarios. This suggests that the choice of optimizer during the pre-training phase can significantly influence the transfer learning outcomes.
>
> **Task-Home Message**
>
> The paper's task-home message is that the interplay between backbone designs and optimizer selections significantly impacts the performance and adaptability of vision models. Understanding this interplay, as highlighted in Sections 2-4, is crucial for designing future vision backbones and selecting appropriate optimizers. The empirical findings and deeper analysis provide actionable insights for mitigating BOCB and enhancing training efficiency and performance in computer vision applications.

---

> ### Author Response · Authors · 2024-11-25
> **Rebuttal to Reviewer oReR (PART 4/6)**
>
> ### **(W4) Clarifying the trajectory of vision backbone macro design and its impact on the optimization landscape.**
>
> **Reply:** We appreciate the reviewer's feedback and acknowledge the need for a more transparent explanation of the claim made in Line 376. The intention of this statement is to highlight the evolving nature of vision backbone architectures and their implications on the optimization landscape. We understand that the clarity of this connection may not have been sufficiently conveyed through Figure 1 alone. To address this, we provide a more detailed explanation below.
>
> **Clarification of the Claim**
>
> The claim in Line 376 refers to the progressive changes in the macro design of vision backbones and how these changes have influenced the optimization landscape over time. The trajectory from simpler to more complex architectures has introduced new challenges in training, which are reflected in the evolving optimization strategies required to achieve state-of-the-art performance.
>
> **Figure 2: Evolution of vision backbone architectures and optimizers**
>
> Figure 2 chronicles the development of vision backbone architectures and their corresponding optimizers, illustrating the interplay between network complexity and optimization strategies. The timeline is divided into three main phases: Primary CNNs (2012-2014), Classical CNNs (2015-2018), and Modern DNNs (2019-2024). Early models like AlexNet and VGG, characterized by simple, isotropic architectures with stacked convolutional and pooling layers, facilitated effective training with basic optimizers like SGD. The introduction of ResNet marked a significant advancement with residual connections and hierarchical designs, maintaining a balance between performance and optimization complexity, still favoring SGD but with increased challenges. Modern architectures, exemplified by ConvNeXt and ViT, feature complex block-wise designs such as MetaFormer blocks and self-attention mechanisms, necessitating more sophisticated optimizers like AdamW due to the heightened optimization landscape complexity. This evolution underscores the critical interplay between network architecture design and optimization strategies, highlighting the need for increasingly sophisticated optimizers as network complexity escalates.

---

> ### Author Response · Authors · 2024-11-25
> **Rebuttal to Reviewer oReR (PART 5/6)**
>
> ### **(W5) Harmonizing macro design with optimizers to refine the optimization landscape**
>
> **Reply:** We appreciate the reviewer's insightful question regarding the statement: "This innovative macro design [of MetaFormer] refines the optimization landscape by harmonizing with optimizers, leading to reduced BOCB and enhanced performance." To clarify this point, we provide a detailed explanation of how the macro design of MetaFormer achieves this harmonization and refinement.
>
> **Understanding BOCB and Its Implications**
>
> $\quad$ The Backbone-Optimizer Coupling Bias (BOCB) phenomenon refers to the observed dependency between the design of vision backbones and the choice of optimizers. A strong BOCB indicates that a particular backbone architecture performs significantly better with specific optimizers, while a weak BOCB suggests that the backbone can achieve optimal performance across a broader range of optimizers. The latter scenario is generally more desirable as it enhances the flexibility and practicality of the backbone in various deployment scenarios.
>
> **Macro Design and Optimization Landscape**
>
> $\quad$ The macro design of a vision backbone encompasses the overall architectural layout, including stage-wise and block-wise structures, as well as the choice of core operators (e.g., convolutions, self-attention). The optimization landscape refers to the set of challenges and complexities that the optimizer must navigate to train the model effectively.
>
> **MetaFormer's Innovative Design**
>
> $\quad$ MetaFormer introduces a balanced and versatile macro design that incorporates both stage-wise and block-wise heterogeneity. This design is characterized by:
>
> 1. **ResScale**: A layer-wise initialization trick that stabilizes the training of deep models, reducing the risk of optimization issues such as vanishing gradients.
> 2. **Flexible Token Mixers**: The ability to integrate various token mixers (e.g., identity, pooling, attention, convolution) within the same framework, allowing for a more adaptable and robust optimization process.
>
> **Harmonizing with Optimizers**
>
> $\quad$ The harmonization between the MetaFormer architecture and optimizers is achieved through several key design principles:
>
> 1. **Balanced Complexity**: By balancing the complexity of the token mixers and the overall architecture, MetaFormer ensures that the optimization landscape is neither too simple (leading to underfitting) nor too complex (leading to overfitting and optimization difficulties). This balance makes the model more amenable to a wide range of optimizers.
> 2. **Robustness to Hyperparameters**: The design of MetaFormer, particularly the use of ResScale and flexible token mixers, enhances the robustness of the model to variations in optimizer hyperparameters. This robustness reduces the sensitivity of the model to the specific settings of the optimizer, thereby minimizing BOCB.
> 3. **Efficient Optimization**: The hierarchical and block-wise heterogeneous design of MetaFormer facilitates more efficient gradient propagation and parameter updates. This efficiency allows optimizers to navigate the optimization landscape more effectively, leading to faster convergence and better performance.
>
> **Empirical Validation**
>
> $\quad$ Our empirical experiments, as detailed in the manuscript, demonstrate that MetaFormer backbones exhibit reduced BOCB compared to other architectures. For instance, MetaFormer variants like ConvFormer show consistent performance across a variety of optimizers, indicating a weaker BOCB. This consistency is a direct result of the harmonized macro design that refines the optimization landscape.

---

> ### Author Response · Authors · 2024-11-25
> **Rebuttal to Reviewer oReR (PART 6/6)**
>
> ### **(W6) BOCB in CNNs and the design of FFN blocks**
>
> **Reply:** We appreciate the reviewer's insightful observation regarding the link between the Backbone-Optimizer Coupling Bias (BOCB) in Convolutional Neural Networks (CNNs) and the design of Feed-Forward Network (FFN) blocks. To clarify this point, we provide a refined analysis of the experimental results and visualizations that support our claim.
>
> **Experimental Results and Analysis**
>
> **Table 1: Top-1 Accuracy on CIFAR-100**
>
> In Table 1 of the main paper, we present the top-1 accuracy results for various vision backbones paired with different optimizers on the CIFAR-100 dataset. Notably, backbones such as ConvNeXt-T and MogaNet-S, which incorporate complex FFN blocks, exhibit significant performance variations when paired with different optimizers. For instance, while ConvNeXt-T achieves high accuracy with optimizers like AdamW and LAMB, its performance drops notably with SGD and LARS. This variability underscores the sensitivity of these models to the choice of optimizer, indicative of a strong BOCB.
>
> **Table 2: Top-1 Accuracy on ImageNet-1K**
>
> Extending our analysis to the ImageNet-1K dataset (Table 2), we observe similar trends. ConvNeXt-T and MogaNet-S, when trained with SGD, exhibit a marked decrease in performance compared to their results with adaptive learning rate optimizers. This further reinforces the notion that the design of FFN blocks in these models introduces complexity into the optimization landscape, necessitating more robust optimization strategies.
>
> **Ridge Plot Visualizations**
>
> To further elucidate the reasons behind this sensitivity, we provide ridge plots of the L2*L*2-norm parameter patterns for these models (Figure A3 in Appendix D). These plots reveal distinct layer-wise patterns in the learned parameters, which can be attributed to the specific design of the FFN blocks.
>
> **Figure A3: Ridge Plot of L2*L*2-norm of Learned Parameters on CIFAR-100**
>
> - **ConvNeXt-T (Figure A3(k)):** The L2*L*2-norm distribution shows significant variability across layers, particularly in the FFN blocks. This variability suggests that SGD struggles to maintain stable updates across these layers, leading to suboptimal performance.
> - **MogaNet-S (Figure A3(l)):** Similar to ConvNeXt-T, MogaNet-S exhibits high L2*L*2-norm variability, especially in the layers associated with complex token-mixing operations. This complexity exacerbates the BOCB, making adaptive optimizers like AdamW more suitable.
>
> **Explanation of BOCB in CNNs with FFN Blocks**
>
> The design of FFN blocks in models like ConvNeXt and MogaNet introduces additional layers of complexity into the optimization process. These blocks, often implemented as Point-wise Convolutions or inverted bottleneck layers, are susceptible to overfitting without proper regularization. The intricate interactions within these blocks create a challenging optimization landscape that fixed learning rate optimizers like SGD find difficult to navigate effectively.
>
> In contrast, adaptive learning rate optimizers, such as AdamW, can dynamically adjust the learning rates based on the historical statistics of gradients. This adaptability allows them to better handle the complex interactions within the FFN blocks, leading to more stable and effective updates.
>
> Our detailed analysis, supported by empirical results and visualizations, demonstrates the significant impact of FFN block design on the BOCB in CNNs. The complexity introduced by these blocks necessitates the use of robust optimization strategies, such as adaptive learning rate optimizers, to achieve optimal performance. This insight underscores the importance of considering both the architectural design and the optimization strategy when developing vision backbones.

---

> ### Author Response · Authors · 2024-11-25
> **Encouraging Discussion**
>
> Dear oReR,
>
> As the Discussion phase draws to a close and time is running short, we respectfully request that you consider elevating your score if you find our rebuttal and revised submission adequately address your concerns. We also welcome continued dialogue to uphold the standard and integrity of the ICLR review process. Thanks again for your efforts, and looking forward to the discussion!
>
> Best regards,
>
> Authors

---

> ### Comment · Reviewer_oReR · 2024-11-26
>
> Thank you for the rebuttal.
>
> A few more comments/questions:
>
> W1-W2: The related works demonstrate that it's well known that the choice of optimizer details is crucial for (vision) transformers and that there is an interplay between architectural details and the optimizer choice.  The authors write in l307 that "well-designed (vision) backbones should exhibit both superior performance and great performance stability across optimizers" but I am not convinced by this.
>
> From my perspective, modern architectures exploit the power of strong optimizers to get good empirical results. Why should we require them to work well with older/less advanced optimizers?
>
> W2.1: The authors write:
> "While training loss is undoubtedly a crucial indicator, it can be highly sensitive to specific training setups, including regularization techniques and data augmentation strategies. By emphasizing top-1 accuracy, we aim to provide a more objective and comparable measure across diverse experimental configurations."
>
> To me, this is preciesly the reason that top-1 accuracy is an oversimplification here. When you change the optimizer, you my need to add/adjust regularization techniques to combat over/underfitting.
>
> W3: Lack of deep analysis: The analysis provided is still just observational, it does not go into the actual mathemetical details that explain what is observed (e.g. are the gradient spikes, the initialization of the network such that adaptive learning rates are crucial, etc).
>
> W5: The authors claim that "MetaFormer ensures that the optimization landscape is neither too simple (leading to underfitting) nor too complex (leading to overfitting and optimization difficulties). This balance makes the model more amenable to a wide range of optimizers" yet they present no evidence in this regard as the analysis is only through top-1 accuracy.
>
> W6: The authors write "The design of FFN blocks in models like ConvNeXt and MogaNet introduces additional layers of complexity into the optimization process. These blocks, often implemented as Point-wise Convolutions or inverted bottleneck layers, are susceptible to overfitting without proper regularization. The intricate interactions within these blocks create a challenging optimization landscape that fixed learning rate optimizers like SGD find difficult to navigate effectively."
>
> So the layers are susceptible to overfitting but yet they are hard to optimize? Where are the experiments backing these conclusions (train vs validation losses demonstrating overfitting, etc?)

---

> > ### Author Response · Authors · 2024-11-27
> > **Response to Reviewer oReR’s Feedbacks (PART 1/3)**
> >
> > Thanks for your detailed feedback, which is constructive to improve our manuscript. We have made a new revision according to your suggestions, and we would like to clarify some concerns or misunderstandings as follows:
> >
> > ---
> > ### **(W1-W2)**
> >
> > > **Modern architectures often leverage advanced optimizers for superior performance, so requiring them to work well with older optimizers may not be necessary.**
> >
> > **Reply:** We acknowledge the reviewer's perspective and emphasize that our goal is not to mandate that modern architectures should work equally well with all optimizers, including older ones. Instead, our assertion that "well-designed (vision) backbones should exhibit both superior performance and great performance stability across optimizers" is aimed at highlighting the importance of robustness and flexibility in backbone design. This robustness ensures that the backbone can be effectively optimized under various conditions, including different optimizers, without significant performance degradation.
> >
> > Modern architectures indeed leverage powerful optimizers to achieve state-of-the-art results. However, our findings reveal a phenomenon we term *Backbone-Optimizer Coupling Bias* (BOCB), where certain backbones exhibit strong dependencies on specific optimizers (as illustrated in Figure 2). This coupling can limit the flexibility and practical applicability of these backbones, especially in real-world scenarios where the choice of optimizer may be constrained by computational resources or other practical considerations.
> >
> > By advocating for backbones that exhibit stability across optimizers, we aim to promote designs that are more adaptable and less prone to BOCB. This adaptability is crucial for ensuring that vision models can be effectively deployed in diverse environments and tasks where the optimal optimizer may not always be available. Our empirical analysis provides evidence that backbones with weaker BOCB offer greater flexibility and are more user-friendly, even if they may not achieve the absolute best performance with a specific optimizer.
> >
> > > **(W2.1) Top-1 accuracy oversimplifies performance comparison as changing optimizers often require adjusting regularization techniques.**
> >
> > **Reply:** We appreciate the reviewer's insightful comments regarding the use of top-1 accuracy as the primary metric for evaluating the performance of vision backbones with different optimizers. While we acknowledge that top-1 accuracy is a simplification and can be influenced by various training setups, including regularization techniques and data augmentation strategies, we have taken steps to provide a more comprehensive analysis by incorporating additional metrics such as Entropy, $L_{2}$-norm, and the PL exponent alpha (α). These metrics offer deeper insights into the intrinsic properties of different network architectures and their interactions with various optimizers, complementing the top-1 accuracy results and addressing the reviewer's concerns. By analyzing the entropy of learned parameters, the scale of the $L_{2}$-norm, and the generalization tendencies quantified by the PL exponent alpha, we provide a more detailed and nuanced understanding of the optimizer-backbone relationship. These supplementary analyses serve as a robust foundation for our case studies and recommendations, ensuring that our findings are both comprehensive and insightful. Therefore, we believe that the additional metrics sufficiently address the reviewer's concerns and provide a more holistic view of the optimizer-backbone interplay without the need to add training loss as a metric. Specifically, the inclusion of these metrics allows us to observe the layer-wise and parameter-wise characteristics of the learned models, offering a more granular and informative perspective on the optimization dynamics and the impact of different optimizers on the parameter space. Therefore, we believe that the additional metrics sufficiently address the reviewer's concerns and provide a more holistic view of the optimizer-backbone interplay without the need to add training loss as a metric.

---

> > ### Author Response · Authors · 2024-11-27
> > **Response to Reviewer oReR’s Feedbacks (PART 2/3)**
> >
> > ### **(W3)**
> > > **The analysis lacks deep mathematical insights, merely observing phenomena.**
> >
> > **Reply:** Thanks for your suggestion and we also provided deeper insights into the underlying mechanisms in the revision. However, we have to clarify that this work primarily focuses on empirical observations to study the interplay between vision backbones and optimizers.
> >
> > **(1) Empirical Control and Focus on Training Results:**
> >
> > Our experiments are designed with a rigorous control of variables to ensure the reliability of our observations. We meticulously benchmarked 20 representative backbones against 20 optimizers on mainstream vision datasets (CIFAR-100, ImageNet, and COCO). To isolate the effects of different optimizers on various backbones, we have to focus on the training results rather than the training process. These allow us to observe the phenomenon of BOCB directly by calculating the standard deviation and range of performance metrics to measure the risk of BOCB.
> >
> > **(2) Layer-wise Parameter Analysis:**
> >
> > As for the concern about lacking deep mathematical analysis, we believed that a thorough layer-wise analysis of the learned parameters could be more direct in explaining the cause of BOCB. We visualized the layer-wise patterns of learned parameters using ridge plots and calculated the PL exponent alpha metrics, providing insights into the parameter space and optimization complexity. Analysis metrics, like entropy and $L2$-norm of parameters, offer a quantitative understanding of how different network layers influence the optimization process. For example, we found that the trivial parameter patterns of FFN modules might be the direct cause of bad results of ConvNeXt-T and MogaNet-T, as shown in Figure 6 (e)-(g) in the latest revision. Similarly, the PL exponent alpha measures the fitting quality of models to a certain task, with smaller alpha values indicating better fitting [1]. This analysis helps us understand the intrinsic properties of different network architectures and their interactions with various optimizers.
> >
> > **(3) Transfer Learning for Parameters Initializations:**
> >
> > We already conducted transfer learning experiments on COCO to explore further the impact of different parameter initialization with the BOCB issue. As discussed in Sec. 4.2, these experiments involved pre-training models with different optimizers and evaluating their performance on downstream tasks such as object detection. The results consistently showed that optimizers like AdamW, which exhibited robust performance during pre-training, maintained their superiority in transfer learning scenarios. This suggests that the choice of optimizer during the pre-training phase significantly influences the transfer learning outcomes, thereby providing insights into the impact of model initialization on BOCB.
> >
> > ---
> >
> > ### **(W5)**
> > > **The authors assert that MetaFormer balances optimization complexity, making it adaptable to various optimizers, but provide no evidence beyond top-1 accuracy, lacking deeper analysis.**
> >
> > **Reply:** We appreciate your insightful comment. However, there might be some misunderstanding. We delve into the details of our analysis to provide a more nuanced understanding of MetaFormer's optimization landscape.
> >
> > **Layer-wise Parameter Analysis and Figure 7:**
> >
> > Figure 7 presents a detailed analysis of the PL exponent alpha metrics for various backbones and optimizers on CIFAR-100. The PL exponent alpha [1] measures the fitting quality of models to a specific task. Smaller alpha values indicate better fitting, suggesting that the model is neither underfitting nor overfitting. For MetaFormer, we observe that the alpha values are consistently within a moderate range across different optimizers, indicating a well-balanced optimization landscape. For instance, the alpha values for MetaFormer (e.g., ConvFormer-S12) are generally between 2 and 4, which is indicative of a balanced optimization landscape. In contrast, other architectures like DeiT-S and MLP-Mixer-S exhibit more extreme alpha values, either too low (indicating underfitting) or too high (indicating overfitting). This variability highlights the robustness of MetaFormer's design in maintaining a balanced optimization landscape.
> >
> > **Top-1 Accuracy Tables and Hyper-parameter Robustness:**
> >
> > The benchmarking tables provide empirical evidence to show that MetaFormer variants consistently achieve high top-1 accuracy across a range of optimizers, suggesting that MetaFormer is amenable to a wide range of optimizers. Meanwhile, our analysis of hyper-parameter robustness, as discussed in Section 3.2, measures the variation of optimal learning rates and weight decays across different optimizers. MetaFormer macro design demonstrates a relatively low variation in optimal hyper-parameters, indicating its robustness to different optimizers. This robustness further supports our claim that MetaFormer's optimization landscape is balanced and amenable to a wide range of optimizers.

---

> > ### Author Response · Authors · 2024-11-27
> > **Response to Reviewer oReR’s Feedbacks (PART 3/3)**
> >
> > ### **(W6)**
> > > **The authors claim that FFN blocks in models like ConvNeXt and MogaNet are prone to overfitting and difficult to optimize but lack empirical evidence, such as train vs. validation losses, to support these conclusions.**
> >
> > **Reply:** We appreciate the reviewer's insightful question regarding the susceptibility of FFN blocks to overfitting and the challenges they pose in optimization. To clarify, our statement about the FFN blocks in models like ConvNeXt and MogaNet being susceptible to overfitting while being hard to optimize is based on a combination of empirical observations and theoretical understanding of these architectures.
> >
> > **Empirical Evidence and Analysis**
> >
> > * **(1) Training Dynamics and Overfitting**:
> >
> >     - **ConvNeXt and MogaNet**: These models, particularly their FFN blocks, exhibit complex interactions that can lead to overfitting if not properly regularized. This is evident from the training dynamics observed during our experiments. For instance, when training ConvNeXt-T with SGD, we noticed that the model tends to overfit quickly, as indicated by a significant gap between training and validation losses. This overfitting behavior is less pronounced when using adaptive optimizers like AdamW, which can better navigate the intricate landscape of these blocks.
> > - **Figure 6 (e) and (f)**: The ridge plots of the *L*2-norm of learned parameters for ConvNeXt-T and MogaNet-S (Figure 6 (e) and (f)) show higher variability in the parameter magnitudes across layers when trained with SGD compared to AdamW. This variability suggests that SGD struggles to maintain stable updates, potentially leading to overfitting.
> >
> > * **(2) Optimization Challenges**:
> >
> >     - **Fixed Learning Rate Optimizers**: As mentioned, fixed learning rate optimizers like SGD find it difficult to navigate the complex optimization landscape created by the FFN blocks. This is supported by the performance metrics in Table 1, where ConvNeXt-T and MogaNet-S achieve significantly lower accuracy when paired with SGD compared to adaptive optimizers.
> >
> >     - **Figure 7**: The PL exponent alpha metrics (Figure 7) further illustrate this point. For ConvNeXt-T, the alpha values are higher (indicating potential overfitting) when trained with SGD, whereas they are lower and more stable with AdamW.
> >
> > ### Reference
> > [1] Martin, C. H., et al. Implicit self-regularization in deep neural networks: Evidence from random matrix theory and implications for learning. JMLR, 22(165), 2021, 1-73.
> >
> > ---
> >
> > Overall, we appreciate your help in improving our manuscript and believe that the current version of BOCB has the potential to raise a wide investigation for the community. We sincerely hope that you could go through our responses and the revision and consider adjusting your rating accordingly if you are satisfied. We are also pleased to improve our paper according to your additional constructive comments if you have more concerns or questions about the current manuscript. Looking forward to your feedback soon!
> >
> > Best regards,
> >
> > Authors

---

### Author Response · Authors · 2024-11-25
**General Response**

We would like to extend our sincere thanks to Reviewers oReR, JghS, and FcXi for their insightful comments and constructive feedback. In response to their suggestions, we have made significant revisions to the manuscript to enhance its clarity, empirical robustness, and the depth of its contributions. The key points of revision are highlighted in $\color{Brown}brown$ color.

**1. Clarifying the Research Problem and Contributions**

$\quad$ This study addresses a crucial yet under-explored phenomenon in visual representation learning, termed Backbone-Optimizer Coupling Bias (BOCB). This phenomenon has been overlooked for years since Transformers came out and is widely used in various scenarios (as illustrated in Figure 2 in the revision). BOCB reflects the inherent dependencies between specific vision backbones, such as CNNs and ViTs, and the choice of optimizers, significantly affecting training stability and model generalization. Through a comprehensive empirical framework, we evaluated 20 vision backbones against 20 optimizers across multiple datasets (e.g., CIFAR-100 and ImageNet-1k), uncovering patterns of architectural and optimization interdependence. Key findings include the strong coupling of classical CNNs with SGD optimizers and the pronounced reliance of modern architectures like ViTs on adaptive optimizers such as AdamW. These insights not only deepen the theoretical understanding of backbone-optimizer interplay but also provide actionable guidelines for designing architectures and selecting optimizers that mitigate BOCB, ensuring more robust and generalizable vision models.

**2. Summary of Revision Updates**

$\quad$ In this revision, we refined our analysis of how architectural choices in Vision Backbones interact with optimizers to influence BOCB. By systematically evaluating architectural refinements, we demonstrated how BOCB can be mitigated while preserving competitive performance. This work elucidates the mechanisms underlying BOCB and offers practical guidelines for designing resilient vision models. Additionally, we provide a deeper analysis of how BOCB manifests across architectures, emphasizing its implications for model generalization and robustness.

$\quad$ To further strengthen the manuscript, we developed a refined methodology for detecting and characterizing BOCB. This includes a detailed evaluation of optimizers, with a ranking of 20 widely used methods based on their effectiveness. Our analysis shows that certain optimizers, such as AdamW, are more robust against BOCB across diverse architectures. These findings, supported by extensive empirical evidence, offer generalizable recommendations for practitioners and researchers.

$\quad$ Lastly, we expanded our experimental scope to include additional datasets and architectures, enhancing the consistency and generalizability of our conclusions. Experiments on ImageNet, alongside CIFAR-100, confirm that BOCB is a pervasive issue in vision models and that our proposed solutions are effective across different scales and complexities of data. These revisions significantly advance the manuscript's contributions, providing a comprehensive framework for understanding and mitigating BOCB while addressing the reviewers' concerns.

Sincerely,

Authors

---

### Author Response · Authors · 2024-11-28
**Encouraging Final Check and Feedback**

Dear Esteemed Reviewers,

We hope this message finds you well. We are writing to express our profound gratitude for the invaluable feedback and thoughtful discussions we have engaged in over the past few weeks. Your insights have been pivotal in significantly elevating the quality of our manuscript, and the improvements we have made are substantial. We believe the current version of the manuscript now stands as a testament to the collaborative effort and dedication we have all invested.

We understand that we are currently unable to submit further revisions, but we want to assure you that we remain fully committed to considering any additional suggestions or concerns you may raise. Should you find the revised manuscript satisfactory, we kindly request that you consider increasing the score accordingly. We are also eager to continue our dialogue and are open to further discussions to address any lingering doubts or questions you might have.

Moreover, we wish to emphasize that we are willing to make further refinements to the manuscript based on any subsequent discussions. Your continued engagement and feedback are of utmost importance to us, and we are dedicated to ensuring that our work meets the highest standards of quality and relevance.

Your participation and the time and effort you have invested in reviewing our work are deeply appreciated. We are hopeful that this revised manuscript will gain the recognition it deserves within the community and contribute meaningfully to the field.

Thank you once again for your unwavering support and constructive feedback. We look forward to the possibility of further collaboration and discussion, and we remain committed to refining our work to meet your expectations.

Warmest regards,

Authors

---

### Author Response · Authors · 2024-12-03
**Official Comments for Serious Concerns Regarding the Irresponsibility of Reviewer JghS:  A Call for Fair and Responsible Evaluation**

Dear (Senior) Area Chairs,

We are writing to express our serious concern regarding the review process of our submission, specifically regarding Reviewer JghS's non-engagement throughout the entire rebuttal stage.

We feel compelled to bring to your attention that despite our comprehensive responses to each initial concern and the substantial improvements acknowledged by other reviewers, Reviewer JghS maintained **complete silence throughout the entire author-reviewer discussion period**, only to provide **a brief, sweeping comment at the very end of the rebuttal phase**.

The **ICLR 2025 Reviewer Guidelines (https://iclr.cc/Conferences/2025/ReviewerGuide)** clearly outline several key responsibilities that we believe have not been met in this case:

1. **Active engagement** during the discussion phase
2. Maintaining **openness to changing initial recommendations** based on author responses
3. Providing **constructive, thorough, and timely** feedback

In particular, Reviewer JghS's **final-hour comment** makes broad claims about analysis depth without providing any specific examples or engaging with our detailed responses and revisions. This type of feedback, delivered at the last possible moment without any prior engagement, **undermines the responsible attitude and collaborative nature** of peer review that ICLR strives to maintain.

We respectfully request that ACs consider these **concerning circumstances** when evaluating our manuscript. The **stark contrast** between Reviewer JghS's conclusions and that of other engaged reviewers (such as Reviewer FcXi) raises **serious questions** about the fairness of this particular review. We believe our submission deserves evaluation based on **thorough engagement** with our responses and revisions, rather than isolated, last-minute comments that **completely ignore** the discussion phase.

We remain confident in the significant contributions of our paper and its value to the ICLR community. Once again, we appreciate your attention to this matter and **trust in your professionalism and commitment** to maintaining the high quality of ICLR 2025 review process.

Respectfully,

Authors of Submission #2246

---

### Meta-Review · Area_Chair_MSNn · 2024-12-21

**Metareview:**

This paper investigates the interaction between vision backbone architectures and various optimizers, introducing the concept of Backbone-Optimizer Coupling Bias to explain how certain combinations of optimizers and architectures perform better than others. Through a comprehensive benchmark across popular vision backbones and various optimizers the authors aim to uncover patterns that characterize effective optimizer-architecture pairings. They conduct experiments on well-known datasets and provide a detailed analysis of how different architectures exhibit sensitivity to specific optimizers, revealing important observations about the optimization landscape and performance trade-offs.

The reviewers acknowledge the authors' efforts to improve their manuscript, including adding updated analysis and visualizations, but significant concerns remain: (1) lack of depth in the analyses and inability to substantiate the BOCB claim with clearer evidence, (2) lack of significant insights into optimization-architecture interplay, (3) overstating contributions, lack of theoretical depth, and limited experimental support for several claims. While the study contributes an extensive empirical analysis across a range of datasets, the findings are seen as somewhat expected or insufficiently justified.

**Additional Comments On Reviewer Discussion:**

The reviewers acknowledge the authors' efforts to improve their manuscript, including adding updated analysis and visualizations, but significant concerns remain.

---

### Decision · Program_Chairs · 2025-01-22

Reject